# Structured Initialization for Vision Transformers

Jianqiao Zheng*    Xueqian Li    Hemanth Saratchandran    Simon Lucey

Australian Institute for Machine Learning
The University of Adelaide

## Abstract

Convolutional Neural Networks (CNNs) inherently encode strong inductive biases, enabling effective generalization on small-scale datasets. In this paper, we propose integrating this inductive bias into ViTs, not through an architectural intervention but solely through initialization. The motivation here is to have a ViT that can enjoy strong CNN-like performance when data assets are small, but can still scale to ViT-like performance as the data expands. Our approach is motivated by our empirical results that random impulse filters can achieve commensurate performance to learned filters within a CNN. We improve upon current ViT initialization strategies, which typically rely on empirical heuristics such as using attention weights from pretrained models or focusing on the distribution of attention weights without enforcing structures. Empirical results demonstrate that our method significantly outperforms standard ViT initialization across numerous small and medium-scale benchmarks, including Food-101, CIFAR-10, CIFAR-100, STL-10, Flowers, and Pets, while maintaining comparative performance on large-scale datasets such as ImageNet-1K. Moreover, our initialization strategy can be easily integrated into various transformer-based architectures such as Swin Transformer and MLP-Mixer with consistent improvements in performance.

## 1 Introduction

Despite their success on large-scale training datasets, Vision Transformers (ViTs) often suffer a notable drop in performance when trained on small-scale datasets. This limitation is primarily attributed to their lack of architectural inductive biases, which are crucial for generalization with insufficient data. In contrast, Convolutional Neural Networks (CNNs) possess strong inductive biases that allow them to perform well even with limited training data.

To bridge this performance gap, several strategies have been proposed. These include self-supervised pretraining on large-scale datasets [8, 26], advanced data augmentation techniques [36, 6], and hybrid architectures that incorporate convolutional layers into Vision Transformers [32, 16, 35, 15, 7]. More recently, Zhang *et al.* [37] explored the use of pretrained weights to initialize ViTs. Building on this idea, subsequent works have shown that carefully designed network initialization strategies can enhance ViT performance on small-scale datasets without modifying the model architecture. In particular, Trockman and Kolter [28] introduced mimetic initialization that replicates the weight distribution of pretrained ViTs. Similarly, Xu *et al.* [33] proposed directly sampling weights from large pretrained models.

While effective, these methods come with three notable limitations: (1) they focus on replicating the distribution of pretrained attention weights rather than structuring attention maps; (2) they rely on access to pretrained models trained on large-scale data, which is often impractical in domain-specific scenarios; and (3) their effectiveness is often tied to specific model architectures.

---

*jianqiao.zheng@adelaide.edu.au. Code is available at `https://github.com/osiriszjq/structured_initialization`

39th Conference on Neural Information Processing Systems (NeurIPS 2025).

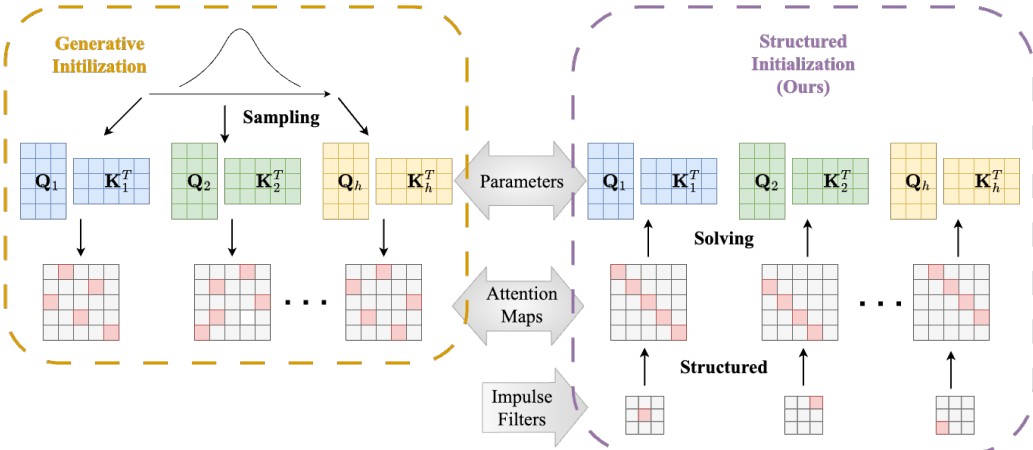

Figure 1: Illustration of conventional generative initialization and structured initialization (ours) strategies for the weights $\mathbf{Q}$ and $\mathbf{K}$ of the attention map in transformers. Conventional generative initialization involves sampling $\mathbf{Q}$ and $\mathbf{K}$ from certain distributions, such as Gaussian or Uniform, resulting in unstructured attention maps. In contrast, our structured initialization imposes constraints on the structure of the initial attention maps, specifically requiring them to be random impulse filters. The initialization of $\mathbf{Q}$ and $\mathbf{K}$ is computed based on this requirement. Note that in both attention maps and random impulse filters, the pink cells indicate ones, while the gray cells represent zeros.

To overcome the limitations of existing approaches, we propose a novel "structured initialization" strategy for ViTs that imposes convolutional structures on attention maps without requiring any pretrained models. Our approach is grounded in our theoretical insights, aligned with the findings of recent work [3], that randomly initialized depthwise convolution filters can match the performance of their trained counterparts in models such as ConvMixer [27] and ResNet [11]. Inspired by this, we develop an initialization strategy for ViTs based on random impulse convolution kernels, which impart locality and structure directly into the attention maps, as shown in Fig. 1. This structured initialization yields inductive biases characteristic of CNNs, enabling ViTs to generalize effectively on small-scale datasets, while preserving their adaptability for large-scale applications. Unlike prior methods that modify ViT architectures by integrating convolutional components, our method preserves the original transformer structure, making it broadly applicable across a variety of ViT variants.

To conclude, our paper makes the following contributions:

- We establish a conceptual link between the structural inductive bias of CNNs and initialization in ViTs, and provide a theoretical justification for using random convolution filters to initialize attention maps.

- To the best of our knowledge, we are the first to introduce the initialization strategy that explicitly structures attention maps in ViTs, embedding convolutional inductive biases without modifying model architecture, enabling compatibility across diverse ViT variants.

- We demonstrate state-of-the-art performance on small-scale and medium-scale datasets, including Food-101, CIFAR-10, CIFAR-100, STL-10, Flowers, and Pets, while maintaining competitive results on large-scale datasets such as ImageNet-1K, and achieving improved performance across various ViT architectures, such as Swin Transformers and MLP-Mixer.

## 2 Related Work

**Introducing inductive bias of CNN to ViT through architecture.** Many efforts have aimed to incorporate a convolutional inductive bias into ViTs through architectural modifications. [7] proposed to combine convolution and self-attention by mixing the convolutional self-attention layers. [21, 15]introduced hybrid models wherein the output of each layer is a summation of convolution and self-attention. [32] explored using convolution for token projections within self-attention, while [35] showed promising results by inserting a depthwise convolution before the self-attention map. [9] introduced gated positional self-attention to imply a soft convolution inductive bias. Although these

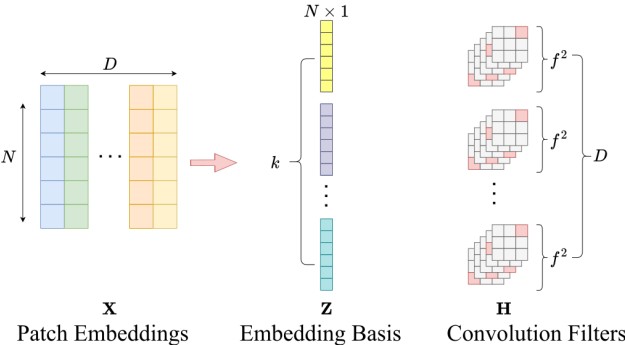

Figure 2: Illustration of why random spatial convolution filters are effective. Patch embeddings $\mathbf{X} \in \mathbb{R}^{N \times D}$ are typically rank-deficient and can be approximately decomposed to $k$ basis. Meanwhile, a linear combination of $f^2$ linearly independent filters $\mathbf{h}$ can express any arbitrary filter in the filter space $\mathbb{R}^{f \times f}$. Based on these two observations, we derive the inequality $D \geq kf^2$ from Proposition 1.

techniques have been proven effective, they aim to introduce the inductive bias of convolution through architectural choices. Our approach, on the other hand, stands out by not requiring any modifications to the architecture, retaining the generalizability to be seamlessly applied to different settings.

**Initializating ViTs from pretrained weights.** The exploration of applying inductive bias through initialization within a transformer is limited to date. [37] posited that the benefit of pretrained models in ViTs can be interpreted as a more effective strategy for initialization. [28, 29] recently investigated the empirical distributions of self-attention weights, learned from large-scale datasets, and proposed a mimetic initialization strategy. While this approach lies between structured and generative initialization, it relies on the pretraining results of large models. [23, 24, 33, 14] directly sampled weights from pretrained large-scale models as initialization for smaller models. While effective, the sampled weights must follow the distribution of these pretrained weights. A key difference in our approach is that our method does not require offline knowledge of pretrained models. Instead, our initialized structure is derived from a theoretical analysis of convolution layers.

**Convolution as attention.** Since their introduction [30, 8], the relationship between transformers and CNNs has been a topic of immense interest. [1] studied the structural similarities between attention and convolution, bridging them into a unified framework. Building on this, [5] demonstrated that self-attention layers can express any convolutional layers through a careful theoretical construction. While these studies highlighted the functional equivalence between self-attention in ViTs and convolutional spatial mixing in CNNs, they did not delve into how the inductive bias of ViTs could be adapted through this theoretical connection. In contrast, our work offers a theoretical insight: a random convolutional impulse filter can be effectively approximated by softmax self-attention.

## 3 Why Random Impulse Filters Work?

It is well established that both ConvMixer and ViTs use alternating blocks of spatial and channel mixing, where ViTs replace the spatial convolutions in ConvNets with attention mechanisms. A fundamental difference, except for the receptive field, lies in their parameterization: spatial convolutions with hundreds of channels typically use distinct kernels for each channel, whereas self-attention relies on a shared mechanism with only a limited number ($\sim 10$) of attention heads. By introducing a key observation that input embeddings are often rank-deficient, we demonstrate that as long as the spatial kernels or attention patterns sufficiently span the kernel space, the uniqueness or repetition of individual kernels becomes less critical. This insight, supported by a recent empirical observation [3], offers a new perspective on the relationship between ViTs and CNNs, motivating the use of impulse-based structures to embed convolutional inductive biases into the attention map initialization.

In recent work [3], Cazenavette *et al*. demonstrated a remarkable performance of randomly initialized convolution filters in ConvMixer and ResNet when solely learning the channel mixing parameters. However, they failed to offer any insights into the underlying reasons. In this section, we provide a theoretical analysis of how solely learning channel mixing can be sufficient for achieving reasonably

good performance. Our theoretical findings are significant as they establish a conceptual link between the architecture of ConvMixer and the initialization of ViT, offering a deeper understanding of desired properties for spatial mixing matrices. Without losing generality, we have omitted activations (*e.g.*, GeLU, ReLU, *etc.*), bias, batch normalization, and skip connections in our equations for clarity.

**Remark 1** *Let us define the patch embeddings or intermediate layer outputs in ConvMixer as* $\mathbf{X} = [\mathbf{x}_1, \mathbf{x}_2, \ldots, \mathbf{x}_D]$, *where* $D$ *is the number of channels and* $N$ *is the number of pixels in the vectorized patch* $\mathbf{x} \in \mathbb{R}^N$. *An interesting observation is the rank (stable rank, defined as* $\sum \sigma^2 / \sigma_{max}^2$*) of* $\mathbf{X}$ *is consistently much smaller than the minimum dimension* $min(N, D)$ *of* $\mathbf{X}$, *indicating a significant amount of redundancy in patch embeddings or intermediate layer outputs in deep networks. This rank deficiency is common in various deep neural networks [10], especially in ViTs [20, 18, 12].*

Let us define a 2D convolution filter as $\mathbf{h} \in \mathbb{R}^{f \times f}$. In general, this kernel can be represented as a circulant matrix $\mathbf{H} \in \mathbb{R}^{N \times N}$, such that $\mathbf{h} * \mathbf{x} = \mathbf{H}\mathbf{x}$, where $*$ denotes convolution operator. The relation between the convolutional matrix and convolution filters is explained in detail in the appendix. A ConvMixer block $\mathbf{T}^{\text{Conv}} : \mathbb{R}^{N \times D} \to \mathbb{R}^{N \times D}$ is composed of a spatial mixing layer $\mathbf{T}_S^{\text{Conv}} : \mathbb{R}^{N \times D} \to \mathbb{R}^{N \times D}$ and a channel mixing layer $\mathbf{T}_C^{\text{Conv}} : \mathbb{R}^{N \times D} \to \mathbb{R}^{N \times D}$, where $\mathbf{T}_S^{\text{Conv}}$ is defined by a sequence of convolution filters $\mathbf{H} = [\mathbf{H}_1, \mathbf{H}_2, \ldots, \mathbf{H}_D] \in \mathbb{R}^{D \times N \times N}$, $\mathbf{H}_i \in \mathbb{R}^{N \times N}$, and $\mathbf{T}_C^{\text{Conv}}$ is defined by a weight matrix $\mathbf{W} \in \mathbb{R}^{D \times D}$. With input $\mathbf{X} = [\mathbf{x}_1, \mathbf{x}_2, \ldots, \mathbf{x}_D] \in \mathbb{R}^{N \times D}$, $\mathbf{T}^{\text{Conv}}$ is

$$\mathbf{T}_S^{\text{Conv}}(\mathbf{X}; \mathbf{H}) = [\mathbf{H}_1\mathbf{x}_1, \mathbf{H}_2\mathbf{x}_2, \ldots, \mathbf{H}_D\mathbf{x}_D], \tag{1}$$

$$\mathbf{T}_C^{\text{Conv}}(\mathbf{X}; \mathbf{W}) = \mathbf{X}\mathbf{W}, \tag{2}$$

$$\mathbf{T}^{\text{Conv}}(\mathbf{X}) = \mathbf{T}_C^{\text{Conv}}(\mathbf{T}_S^{\text{Conv}}(\mathbf{X}; \mathbf{H}); \mathbf{W}) = [\mathbf{H}_1\mathbf{x}_1, \mathbf{H}_2\mathbf{x}_2, \ldots, \mathbf{H}_D\mathbf{x}_D]\mathbf{W}. \tag{3}$$

**Definition 1** *(M–k Spanned Set): Let* $\mathcal{V} = \{\mathbf{v}_1, \mathbf{v}_2, \ldots, \mathbf{v}_N\} \subset \mathbb{R}^d$ *be a finite set of vectors. We say that* $\mathcal{V}$ *is* $M-k$ *spanned (on* $W$*) if there exists a partition of* $\mathcal{V}$ *into at least* $k$ *non-overlapping subsets:*

$$\mathcal{V} = \mathcal{V}_1 \cup \mathcal{V}_2 \cup \ldots \cup \mathcal{V}_k, \tag{4}$$

*such that there exist a* $M-$*dimensional subspace* $W \subset \mathbb{R}^d$ *in the span of each subset* $\mathcal{V}_i$

$$W \subseteq Span(\mathcal{V}_i) \quad \forall i = 1, 2, \ldots, k. \tag{5}$$

**Proposition 1** *A ConvMixer block* $\mathbf{T}$ *consists of a spatial mixing layer* $\mathbf{T}_S(\,\cdot\,; \mathbf{H})$ *with convolution filters* $\mathbf{H}$ *and a channel mixing layer* $\mathbf{T}_C$. $\mathbf{T}'$ *is another ConvMixer block composed of* $\mathbf{T}'_S(\,\cdot\,; \mathbf{H}')$ *and* $\mathbf{T}'_C$. *Let* $k$ *be the rank of input* $\mathbf{X}$. *If* $\mathbf{H}$ *is* $M-k$ *spanned on* $W$, *then for any set of filters* $\mathbf{H}' \subset W$ *and any* $\mathbf{T}'_C$, *there always exists a* $\mathbf{T}_C$ *such that* $\mathbf{T}(\mathbf{X}) = \mathbf{T}'(\mathbf{X})$.

For simplicity, we include the full proof in the appendix. Note that since $H$ are convolution matrices, their span lies in $\mathbb{R}^{f \times f}$ instead of $\mathbb{R}^{N \times N}$, where $f$ is the kernel size. In practice, the fact that $D \geq k f^2$ indicates that randomly initialized spatial convolution kernels are $f^2-k$ spanned and satisfy Proposition 1, as illustrated in Fig. 2. Consequently, any trained results of $\mathbf{T}'_C$ and $\mathbf{T}'_S$ can be achieved by solely training the $\mathbf{T}_C$, while keeping a fixed spatial mixing layer $\mathbf{T}_S$. Hence, the following corollaries can be obtained, and Corollary 1 explains the phenomenon found in [3], mentioned at the beginning of the section. Corollary 2 inspires our proposed structure initialization for ViTs. The related experimental evidence of these corollaries and our proposition is given in the appendix.

**Corollary 1** *Randomly initialized spatial convolution filters perform as well as trained spatial convolution filters since the* $f^2-k$ *spanned condition in Proposition 1 is satisfied.*

**Corollary 2** *Random impulse spatial convolution filters perform as well as trained spatial convolution filters since the* $f^2-k$ *spanned condition in Proposition 1 is satisfied.*

**Corollary 3** *Spatial convolution filters with all ones (referred to as "box" filters) will not perform well since they are* $1-k$ *spanned and can produce only averaging values.*

## 4 Structured Initialization for Attention Map

### 4.1 Expected Initialized Attention Map Structure

ConvMixer and ViT share most of the components in their architectures. The gap in their performance on small-scale datasets stems from their architectural choices regarding the spatial mixing matrix.

Although depthwise convolution (ConvMixer) and multi-head self-attention (ViT) may appear distinct at first glance, their underlying goal remains the same: to identify spatial patterns indicated by the spatial mixing matrix. As defined in Sec. 3, similar to the spatial mixing step in ConvMixer defined in Eq. (1), the spatial mixing step of multi-head attention can be expressed as

$$\mathbf{T}_S^{\text{ViT}}(\mathbf{X}; \mathbf{M}) = [\mathbf{M}_1\mathbf{x}_1, \dots, \mathbf{M}_1\mathbf{x}_d, \mathbf{M}_2\mathbf{x}_{d+1}, \dots, \mathbf{M}_2\mathbf{x}_{2d},$$
$$\dots, \mathbf{M}_h\mathbf{x}_{(h-1)d+1}, \dots, \mathbf{M}_h\mathbf{x}_{h*d}], \tag{6}$$

where $d$ represents the feature dimension in each head, typically set to $D/h$, with $h$ being the number of heads, and the matrices $\mathbf{M}_i$ for multi-head self-attention can be expressed as follows:

$$\mathbf{M}_i = \text{softmax}(\mathbf{X}\mathbf{Q}_i \mathbf{K}_i^T\mathbf{X}^T), \tag{7}$$

where $\mathbf{Q}_i, \mathbf{K}_i \in \mathbb{R}^{D \times d}$ are attention weight matrices.

It is worth noting that in Eq. (1), the spatial matrices $\mathbf{H}$ are in convolutional structure, resulting in a span of $\mathbb{R}^{f \times f}$ instead of $\mathbb{R}^{N \times N}$, despite each $\mathbf{H}_i$ having a size of $N \times N$. This structural constraint ensures that CNNs focus on local features but struggle to capture long-range dependencies. In contrast, the span of spatial matrices $\mathbf{M}$ in Eq. (6) is $\mathbb{R}^{N \times N}$, allowing for greater learning capacity without these limitations. However, a randomly initialized $\mathbf{Q}$ and $\mathbf{K}$ contain no structural information, resulting in random matrices as depicted in the bottom left of Fig. 1.

Leveraging this insight, we propose to initialize the attention map for each head in ViT to a convolutional structure as denoted in the bottom right of Fig. 1. Our initialization strategy preserves both the advantage of locality and the capacity to learn long-range information. For clarity and brevity, the following discussions will focus only on one head of multi-head self-attention. Therefore, from Eq. (6) and Eq. (1), our structured initialization strategy can be represented as

$$\mathbf{T}_S^{\text{ViT}}(\mathbf{X}; \mathbf{M}) \xleftarrow{\text{init}} \mathbf{T}_S^{\text{Conv}}(\mathbf{X}; \mathbf{M}) \Rightarrow \mathbf{M}_{\text{init}} = \text{softmax}(\mathbf{X}\mathbf{Q}_{\text{init}}\mathbf{K}_{\text{init}}^T\mathbf{X}^T) \approx \mathbf{H}. \tag{8}$$

**Why using impulse filters?** Any random convolution filters that satisfy proposition 1 could be a choice of initialization of attention maps to introduce inductive bias. However, random convolution filters $\mathbf{H}$ usually contain both positive and negative values, while the output of the softmax function is always positive, making Eq. (8) unreachable. One straightforward option is to use random positive convolution filters with a normalized sum of one, following the property of softmax. However, this approach often proves inefficient as the patterns are too complicated for a softmax function to handle with $\mathbf{QK}$ being of low rank. In [25], the authors found that the softmax attention map serves as a feature selection function, which typically tends to select a single related feature. In convolution filters, this selection can be parameterized as impulse filters. According to Corollary 2, random impulse filters are also $f^2$–$k$ spanned. In conclusion, when initializing a softmax attention map, the most straightforward and suitable choice is random impulse convolution filters.

**Pseudo input.** The advantage of self-attention is that its spatial mixing map is learned from data. The real input to an attention layer is $\mathbf{P} + \mathbf{X}$ for the first layer and $\mathbf{X}$ (the intermediate output from the previous layer) for the following layers. However, during initialization, there is no prior information about the input. To address this problem, we simply use the initialization of positional encoding $\mathbf{P}$ as the pseudo input in the initialization of $\mathbf{Q}$ and $\mathbf{K}$, replacing the actual input data $\mathbf{P} + \mathbf{X}$ or intermediate outputs. Remember that this only happens when we solve the initialization to avoid data-dependent initialization, while in the training stage, we make no change to the ViT architecture.

With the use of impulse filters and the pseudo input, Eq. (8) becomes

$$\mathbf{M}_{\text{init}} = \text{softmax}(\mathbf{P}\mathbf{Q}_{\text{init}}\mathbf{K}_{\text{init}}^T\mathbf{P}^T) \approx \mathbf{H}_{\text{impulse}}. \tag{9}$$

## 4.2 Solving $\mathbf{Q}_{\text{init}}$ and $\mathbf{K}_{\text{init}}$

There exist numerous approaches to solve Eq. (9) for $\mathbf{Q}_{\text{init}}$ and $\mathbf{K}_{\text{init}}$ with known $\mathbf{H}_{\text{impulse}}$ and $\mathbf{P}$. Here, we apply an SVD-based method. First, we change Eq. (9) to exclude the Softmax function as

$$\mathbf{P}\mathbf{Q}_{\text{init}}\mathbf{K}_{\text{init}}^T\mathbf{P}^T = \alpha\mathbf{H}_{\text{impulse}} + \beta\mathbf{Z}, \tag{10}$$

where $\mathbf{Z} \sim \mathcal{N}(0, \frac{1}{D}\mathbf{I})$. Then to solve for $\mathbf{Q}_{\text{init}}$ and $\mathbf{K}_{\text{init}}$, we put $\mathbf{P}$ to the right-hand side of the equation using the pseudo inverse $\mathbf{P}_{\text{inv}} = (\mathbf{P}^T\mathbf{P})^{-1}\mathbf{P}^T$—since $\mathbf{P}$ is randomly initialized, it will be

of full rank. Therefore, we get

$$\mathbf{Q}_{\text{init}}\mathbf{K}_{\text{init}}^T = \mathbf{P}_{\text{inv}}(\alpha\mathbf{H}_{\text{impulse}} + \beta\mathbf{Z})\mathbf{P}_{\text{inv}}^T . \tag{11}$$

With Eq. (11), we do SVD on the right-hand side and get a low-rank approximation of $\mathbf{Q}_{\text{init}}$ and $\mathbf{K}_{\text{init}}$:

$$\mathbf{U}, \mathbf{s}, \mathbf{V}^T = \text{SVD}\left(\mathbf{P}_{\text{inv}}\left(\alpha\mathbf{H}_{\text{impulse}} + \beta\mathbf{Z}\right)\mathbf{P}_{\text{inv}}^T\right) , \tag{12}$$

$$\widetilde{\mathbf{Q}}_{\text{init}} = \mathbf{U}\sqrt{\mathbf{s}}\left[:,:d\right], \quad \widetilde{\mathbf{K}}_{\text{init}} = \mathbf{V}\sqrt{\mathbf{s}}\left[:,:d\right] . \tag{13}$$

Finally, we do a normalization to get the final $\mathbf{Q}_{\text{init}}$ and $\mathbf{K}_{\text{init}}$ as

$$\mathbf{Q}_{\text{init}} = \frac{\gamma}{\|\widetilde{\mathbf{Q}}_{\text{init}}\|_F)}\widetilde{\mathbf{Q}}_{\text{init}}, \quad \mathbf{K}_{\text{init}} = \frac{\gamma}{\|\widetilde{\mathbf{K}}_{\text{init}}\|_F}\widetilde{\mathbf{K}}_{\text{init}} . \tag{14}$$

In ViT models with $d = 64$, we use a $3 \times 3$ convolution filters by setting $f = 3$. We find that the random values in $\mathbf{Z}$ are not decisive for our initialization strategy. To ensure a clear impulse structure, the initialization is primarily governed by selecting an appropriate ratio of $\alpha$ to $\beta$. Empirically, we choose a large ratio such that $\alpha{:}\beta = 40{:}1$ with $\gamma = 2$. Notably, the exact values of $\alpha$ and $\beta$ are not strictly constrained due to the normalization step, which scales these parameters. The pseudo code for our initialization strategy can be found in the appendix.

## 5 Experiments and Analysis

In this section, we show that our impulse initialization can improve the performance of ViT on small-scale datasets in Sec. 5.1, while it does not limit the learning flexibility of ViT models on large-scale datasets in Sec. 5.2. In Sec. 5.6, we show that even for a pretrained method like weight selection [33], the pretrained model initialized with our impulse structure can provide better pretrained weights with a relatively smaller-scale dataset. Finally, in Sec. 5.4, we show that besides ViTs, the concept of introducing architectural bias into initializations can also be generalized to other models like Swin Transformers and MLP-Mixer. Note that all experiments were conducted on a single node with 8 Tesla V100 SXM3 GPUs, each with 32GB of memory, if not specified. Specifically, all the experiments on the small-scale datasets took about three hours to train each model, while the experiments on the ImageNet-1K took about two days. Please note that all results were reported based on our experiments with retrained models. As a result, minor discrepancies may exist between our reported results and those in published papers. However, the key focus of this analysis remains on the improvements achieved through different initialization strategies.

### 5.1 Small and Medium-Scale Datasets

In this section, we compare the performance of ViT-Tiny under three initialization strategies: the default [31], mimetic [28], and our impulse initialization. Experiments are conducted on medium-scale datasets ($\sim$50K training images), including Food-101 [2], CIFAR-10, and CIFAR-100 [13], as well as small-scale datasets ($\sim$5K training images), such as STL-10 [4], Flowers [19], and Pets [22]. We follow the training recipe from [33], which is proven to be useful in training ViT-Tiny on small and medium-scale datasets. Their codes are based on the timm library [31]. By default, all the weights are initialized with a truncated normal distribution. For a fair comparison, all the experiments were run with identical codes except for the choice of initialization methods. Please also note that although [33] initializes the model weights from large pretrained models, we did not adopt any pretraining step for this experiment. In contrast, we apply default, mimetic, and our impulse initialization methods to ViT-Tiny models and training these models from scratch.

The experimental results are presented in Tab. 1. For simplicity, we keep the statistical results on using different seeds with stochastic filter generations with 5 different runs in Appendix F and Fig. 6. Our method consistently yields substantial improvements over the default initialization across all evaluated datasets. On medium-scale datasets, it achieves performance gains of $2\%{\sim}5\%$, while on small-scale datasets, improvements can reach approximately $8\%$, and in some cases, up to $20\%$. While mimetic initialization also improves the performance, our impulse initialization shows superior efficacy, attributed to the convolutional structure integrated in the attention initialization. Notably, as the dataset size decreases, the performance gap between our method and the default (or mimetic) initialization gets larger. This observation validates that the convolutional inductive bias introduced by our initialization becomes increasingly important when there is less data for the model to learn the spatial dependencies.

Table 1: Classification accuracy of ViT-Tiny with different initialization methods on different datasets. **Green** number indicates an increase in accuracy. Note that we compare the performance to the default initialization method (shaded in gray). ● represents small-scale datasets, and ▲ represents medium-scale datasets. The datasets are ranked based on their training scales.

| Method \ Data↓ | ▲ Food-101 | | ▲ CIFAR-10 | | ▲ CIFAR-100 | | ● STL-10 | | ● Flowers | | ● Pets | |
|---|---|---|---|---|---|---|---|---|---|---|---|---|
| Default [31] | 77.95 | | 92.29 | | 71.67 | | 61.86 | | 64.60 | | 26.58 | |
| Mimetic [28] | 81.78 | 3.83↑ | 93.50 | 1.21↑ | 75.16 | 3.49↑ | 68.54 | 6.68↑ | 71.62 | 7.02↑ | 47.63 | 21.05↑ |
| Ours (impulse) | **81.85** | **3.90↑** | **94.67** | **2.38↑** | **77.02** | **5.35↑** | **70.21** | **8.35↑** | **73.18** | **8.58↑** | **50.84** | **24.26↑** |

Table 2: Classification accuracy of ViT-Tiny, ViT-Small and ViT-Base on ImageNet-1K dataset with different initialization methods. ■ indicates large-scale datasets. Please note that for the last column (shaded in yellow), we report the experimental results on ViT-Base with specific settings to make the default initialization-based ViT comparable to the results reported in concurrent papers.

| Method \ Model | ■ ViT-Tiny | | ■ ViT-Small | | ■ ViT-Base | | ■ ViT-Base* | |
|---|---|---|---|---|---|---|---|---|
| Default [31] | 72.71 | | 79.68 | | 81.24 | | 81.89 | |
| Mimetic [28] | 72.90 | 0.19↑ | 80.26 | 0.58↑ | 80.56 | 0.68↓ | 80.56 | 1.33↓ |
| Ours (impulse) | **72.76** | **0.05↑** | **80.40** | **0.72↑** | **81.83** | **0.59↑** | **82.13** | **0.24↑** |

## 5.2 Large-Scale Datasets

In this section, we compare the performance of ViT-Tiny, ViT-Small, and ViT-base with default, mimetic, and our impulse initialization on a large-scale dataset—ImageNet-1K (over 1M training images). We follow the training recipe from DeiT [26]—a classic and efficient training recipe for training ViT models on ImageNet-1K. We directly use the original ViT structure and training codes in the timm library, except for adding our implementation of initialization. All the models were trained with the same hyperparameters starting from scratch without any pretraining or distillation.

We show the comparison results on the ImageNet-1K dataset in Tab. 2. Detailed training hyperparameters and training curves can be found in Appendix E. In particular, we find that the rapid update for the baseline ViT codebase (*i.e.*, timm library) and the difference in GPU hardware settings result in a small discrepancy in the default initialization-based ViT-Base model between our main result (∼81.24) and the results reported in concurrent papers (∼81.89). Therefore, we have included an additional column (shaded in yellow) in Tab. 2 for ViT-Base* model that uses 16 Tesla V100 GPUs with an extra 0.3 color jittering data augmentation for a clearer comparison.

Despite different training settings and comparisons, our method maintains comparable performance with default initialization, demonstrating that the convolutional inductive bias introduced during initialization does not hinder the model's flexibility in learning data-driven dependencies. This indicates that while the transformer architecture begins training with structurally imposed spatial priors, the attention mechanism retains full capacity to learn optimal feature representations when sufficient training data is available. Furthermore, the convolutional structure we introduced in the attention map initialization not only accelerates early convergence but also improves the robustness to variations in training hyperparameters. We provide more quantitative results in Appendix E.

## 5.3 Training Curves and Analysis

In Fig. 3, we show the training accuracy curves across 300 epochs for different initialization methods of ViT-Tiny on CIFAR-10 and ViT-Base on ImageNet-1K. For the small model on medium-scale dataset, our impulse initialization consistently outperforms the default or mimetic initialization throughout the entire training process. On the large-scale dataset with training large ViT-Base model, our impulse initialization method and the mimetic initialization have shown faster convergence rate than the default initialization at the beginning of the training. However, the mimetic initialization shows degraded performance even to the default initialization at the last 100 training epochs, indicating limited ability in large-scale model training. On the contrary, our method does not limit the learning ability of the large-scale ViT models, showing a significant advantage in the final training stage where the performance surpass both the default and the mimetic initialization methods.

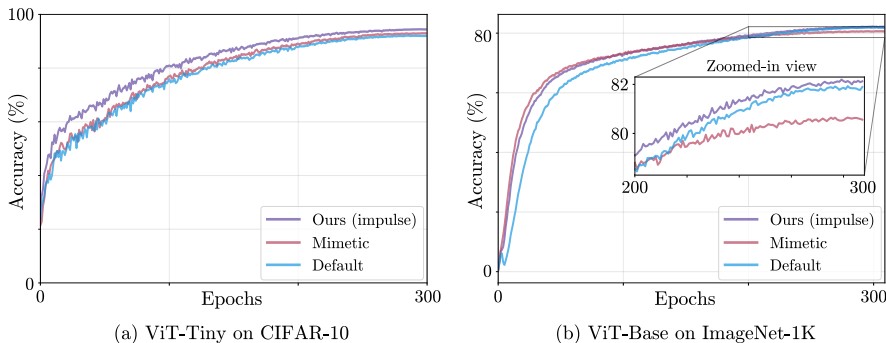

(a) ViT-Tiny on CIFAR-10         (b) ViT-Base on ImageNet-1K

Figure 3: Training curves of ViT-Tiny on CIFAR-10 and ViT-Base on ImageNet-1K using default, mimetic, and impulse initialization. The zoomed-in box shows the training curve in the final training stage from epoch 200 to epoch 300.

Table 3: Classification accuracy of different Swin Transformers and MLP-Mixer on different datasets with different initialization methods. **Red**/**green** number indicates accuracy decrease/increase. We compare the performance to the default initialization method (shaded in **gray**).

| Model&Data
Method | Swin-B on
■ ImageNet-1K | | Swin-T on
▲ CIFAR-10 | | MLP-Mixer on
▲ CIFAR-10 | |
|---|---|---|---|---|---|---|
| Default [31] | 83.14 | | 89.85 | | 87.00 | |
| Mimetic [28] | 83.14 | — | 89.24 | 0.61↓ | — | |
| Ours (impulse) | **83.55** | **0.41↑** | **91.19** | **1.34↑** | **88.78** | **1.78↑** |

In summary, experiments on both small-scale and large-scale datasets show that our initialization method effectively balancing prior architectural knowledge integration with data adaptability—incorporating a convolutional inductive bias during initialization provides structural guidance to capture dependencies when training data is limited, while still allowing the attention mechanism to retain full learning flexibility in scenarios with abundant data.

## 5.4 ViT Variants

Beyond the original ViT architectures, we extend a simplified version of our initialization strategy to the Swin Transformer. Notably, Swin Transformers incorporate relative positional encoding within each attention block—a design choice that aligns with our core objective of instilling convolutional structure into attention maps. For these models, we achieve this by directly initializing the relative positional embeddings with our impulse pattern.

We evaluated our approach with Swin Transformer-Base (Swin-B) architecture on the ImageNet-1K following the training recipe in the original Swin Transformer paper [17]. To further demonstrate the flexibility of our approach, we applied our impulse initialization to the MLP-Mixer. For experiments with Swin Transformer-Tiny (Swin-T) and MLP-Mixer, we follow training settings in [34], which is specifically designed for training different models on CIFAR-10.

The results are summarized in Tab. 3. For models with strong learning capacity but limited inherent inductive biases at initialization, introducing a structured initialization consistently enhances performance without compromising the model's capacity to learn complex data dependencies. In particular, while Swin Transformer incorporates convolutional inductive bias through windowed self-attention, applying our structured initialization further improves the performance by $1.34\%$ on smaller-scale datasets such as CIFAR-10, with no degradation on large-scale datasets like ImageNet-1K. MLP-Mixer, which replaces the spatial convolutions in ConvMixer with MLP layers, typically struggles to train effectively on small-scale datasets. However, initializing the spatial MLPs with a convolutional structure leads to a $1.78\%$ performance gain on CIFAR-10. In contrast, mimetic initialization—designed to replicate empirical weight distributions from pretrained ViTs—shows negligible benefits or degrades performance, highlighting its limited generalizability outside the specific pretrained ViT structures.

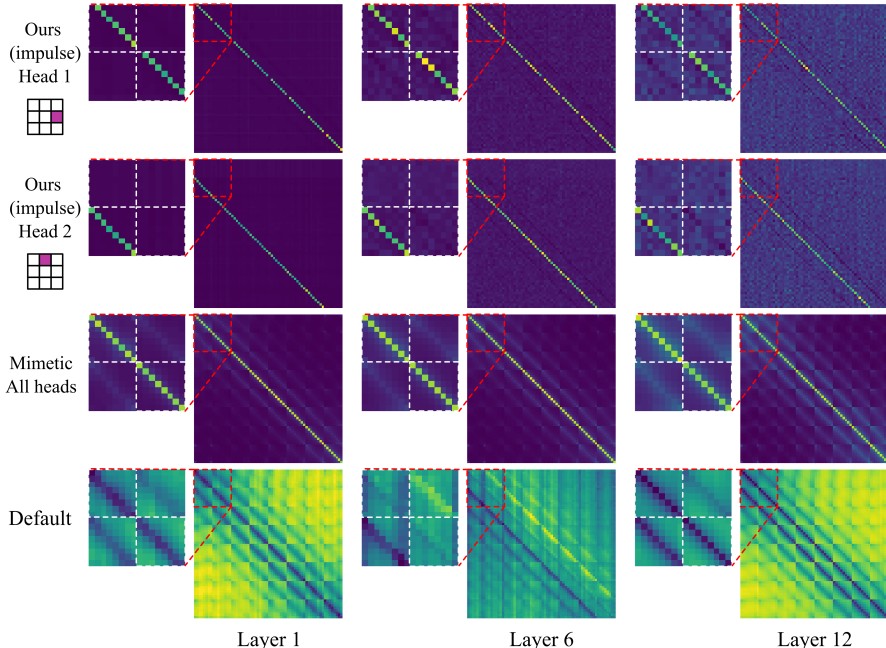

Figure 4: Visualization of attention maps in ViT-T using ours, mimetic [28], and default [33] initializations. Red boxes highlight zoomed-in details of the $16 \times 16$ upper left corner in attention maps. White boxes indicate the $8 \times 8$ sub-blocks of the zoomed-in attention maps. Our structured initialization method offers distinct attention peaks aligned with the impulse structures across different heads. Head 1 offers a peak at $+1$ offset from the main diagonal. Head 2 offers a peek at $-8$ offset (equivalent to $-$image_size) from the main diagonal. Both mimetic and random initialization methods initialize all the attention heads identically. Specifically, mimetic initialization primarily strengthens the main diagonal of the attention map for each head, while random initialization shows minimal structural patterns with flatter peak values.

## 5.5 Attention Maps

The initialized attention maps produced by default, mimetic, and our impulse initializations are shown in Fig. 4. For better visualization, we use $32 \times 32$ CIFAR-10 images as input with a $4 \times 4$ patch size, yielding the token size as $8 \times 8 = 64$. We observe that the default initialization generates near-identical attention values with little distinguishable spatial structure, while the mimetic initialization introduces diagonally dominant values by adding an identity matrix during initialization. Notably, these two initialization methods initialize all the attention heads in the same way. In contrast, our initialization assigns different impulse patterns to each attention head, producing spatially diverse activations with off-diagonal peaks on specific impulse positions.

## 5.6 Pre-trained Model

In this section, we demonstrate that our initialization method remains compatible with existing pretraining pipelines. Building on the work of Xu *et al*. [33], who showed that weight selection from ImageNet-21K-pretrained ViT-Small models effectively initializes ViT-Tiny architectures, we extend this approach to ImageNet-1K pertaining. We pretrained three ViT-Small models on ImageNet-1K using three initialization strategies: default, mimetic, and our proposed impulse initialization. From these, we derived three ViT-Tiny initialization variants—termed "1K-default", "1K-mimetic", and "1K-impulse"—using the same weight selection methodology on smaller-scale datasets. When training these initialized ViT-Tiny models on small and medium-scale datasets (results shown in Tab. 4), we observed that: (1) Switching from ImageNet-21K to ImageNet-1K—less training data—pretraining with default initialization typically incurs a performance drop of ∼1%. (2) Mimetic initialization fails to mitigate this data degradation, (3) The ImageNet-1K pretraining model with our impulse-based

Table 4: Classification accuracy of ViT-Tiny on small-scale datasets with the weight selection method. Here, the weights selected for the experiments were pretrained on the ImageNet-1K (shortened as 1K, shown above the **purple** dashed line) and ImageNet-21K (shortened as 21K, shown below the **purple** dashed line) datasets with different initialization methods. Please note that pretraining on ImageNet-21K with default initialization is the original weight selection method [33]. We compare the performance to the default initialization method pretrained on ImageNet-1K (shaded in **gray**).

| Data↓
Method+Model | ▲ Food-101 | | ▲ CIFAR-10 | | ▲ CIFAR-100 | | ● STL-10 | |
|---|---|---|---|---|---|---|---|---|
| 1K+Default [33] | 85.42 | | 96.61 | | 79.64 | | 82.58 | |
| 1K+Mimetic [28] | 86.32 | 0.90↑ | 96.37 | 0.24↓ | 79.86 | 0.22↑ | 82.39 | 0.19↓ |
| 1K+Ours (impulse) | **87.43** | **2.01↑** | **97.19** | **0.58↑** | 80.92 | 1.28↑ | **83.89** | **1.31↑** |
| 21K+Default [33] | 87.14 | 1.72↑ | 97.07 | 0.46↑ | **81.07** | **1.43↑** | 83.23 | 0.65↑ |

initialization achieves performance parity with ImageNet-21K pretrained baselines. This highlights the robustness of our method even under reduced pretraining data regimes.

## 6 Limitations and Broader Impacts

There exist several limitations in our initialization method: **(1) Positional encoding.** Although positional encoding is a natural and effective choice for pseudo-inputs—due to its simplicity and data-independent nature—even simpler alternatives may exist for initializing the $\mathbf{Q}$ and $\mathbf{K}$ matrices. In this work, we focus solely on the initialization of $\mathbf{Q}$ and $\mathbf{K}$, but a more comprehensive initialization strategy that also considers patch embeddings and positional encodings could offer greater control over the structure of attention maps. Notably, since positional encoding is only added at the input layer, the influence on attention maps may diminish with increasing network depth, as illustrated in Fig. 4. **(2) Hard constraints.** Our Corollary 2 of Proposition 1 is based on the presumption that the filters are $f^2-k$ spanned, which is usually a characteristic inherent in CNNs. However, in ViTs, the limited number of heads may be inadequate to span the filter space of a small kernel. Finding better adaptations in this scenario remains a challenge. **(3) Value initialization.** Our method does not consider the initialization for the value weights $\mathbf{V}$ and the projection matrix.

**Broader impacts.** This work advances the understanding of how structured initialization influences the performance of transformers, particularly in resource-constrained settings. Our research findings enables more efficient training of neural networks on small-scale datasets, which may benefit domains such as medical imaging, environmental monitoring, robotics, or education, where data is limited or expensive to collect. Furthermore, improving initialization strategies can reduce computational costs, contributing to more sustainable AI practices.

However, as with most advances in artificial intelligence, these techniques carry a risk of misuse or harmful societal applications. For example, by applying our initialization method to more powerful models with fewer resources could make larger models more easily accessible for malicious purposes. These concerns further remind us to carefully consider the broader societal impacts of our research and make sure its benefits outweigh potential harms.

## 7 Conclusion

In this paper, we propose a structured initialization method with convolutional impulse filters for attention maps in ViTs. Our method preserves both the advantage of locality within CNNs and the capacity to learn long-range dependencies inherited from ViTs. We also provide a thorough theoretical explanation of the spatial and channel mixing in ConvMixer and ViT, building connections between the structural bias in CNNs and the initialization of ViTs. Our results on small-scale datasets validate the effectiveness of the convolutional structural bias, while on-par performance on large-scale datasets indicates the preservation of architectural flexibility. Our initialization also accelerates early-stage convergence but also enhances the model's robustness to variations in training hyperparameters. Furthermore, we demonstrate that our method consistently provides benefits across a range of architectures and even under pre-training strategies.

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

# Appendix: Structured Initialization for Vision Transformers

## A    Frequently Asked Questions

**Q: Does introducing CNN structural biases to Transformer contradict the advantage of its structure?**

**A:** Our method focuses on introducing the architectural inductive bias from CNNs to initialize the attention map without changing the Transformer architectures. We have emphasized this argument in the abstract and introduction, stating that unlike previous arts [35, 15, 7, 32, 16] that directly introduce convolutions into attention, potentially damaging the structural advantages of Transformers, our method only introduces architectural inductive bias in attention map initialization, maintains the inherent structural flexibility in ViTs. Therefore, our method preserves the Transformer architectures and still allows the ViTs to learn flexible, dynamic global relationships. This is also one of the central innovations of our method. In addition, we have also provided experiments on large-scale applications to validate the stable performance of our method that preserves the architectural flexibility of ViTs.

**Q: Why is the proposed method better than transfer learning on large-scale pretrained models or a hybrid architecture combining CNN and attention?**

**A:** As stated in Sec. 1, we would like to emphasize the advantages of our proposed structured initialization method: 1) Unlike previous methods that do transfer learning on large-scale pre-trained ViTs, our method involves no pre-training of large-scale models on large-scale datasets, which may not be readily available; 2) Our method shares the advantages of both CNNs and ViTs.

In general, introducing inductive bias in CNNs for initializing the attention map helps ViTs begin learning from a more reasonable/stable starting point, while the default random initialization can lead to a noisy starting point, especially when training on small-scale datasets. Notably, our initialization method does not alter the ViT structure, maintaining the advantages of the Transformer architectures in learning dynamic, long-dependent global features. In contrast, methods that directly combine the architectures of CNNs and ViTs alter the Transformer architectures, potentially compromising its architectural advantages. Please also refer to the theoretical analysis of random filters in Sec. 3 and the convolutional representation matrix in Appendix C of the main paper for more theoretical explanations.

For pretrained models, we noted in the introduction section that their reliance on access to pretrained models makes them impractical for domain-specific scenarios. For example, pretraining a large model on extensive datasets only to deploy a smaller model on limited data is inefficient and often unreasonable. Furthermore, their effectiveness heavily depends on the performance of the specific model architectures, offering no inherent advantages from using pretrained weights. Nevertheless, even on pretrained models, our initialization still outperforms other methods. For additional quantitative evidence, please refer to the experimental analysis in Sec. 5.1 and Sec. 5.6.

**Q: Why use an impulse filter? Does enforcing a strict structure in initialization degrade performance?**

**A:** While one might expect that a rigid initialization could limit the model's flexibility, our experimental results in Sec. 5 demonstrate that the impulse structure does not hurt the training. On the contrary, it consistently improves the performance, particularly on small and medium-scale datasets. This is attributed to the structured initialization introducing a beneficial inductive bias, which guides the model toward learning useful representations in the early stage of training.

Moreover, the hyperparameters $\alpha$, $\beta$, and $\gamma$ control the norms of $\mathbf{Q}$ and $\mathbf{K}$, ensuring that the attention maps exhibit a well-defined convolutional structure at initialization, while maintaining sufficiently flexible to adapt and learn from data during training. This design shows a central innovation of our method, and its effectiveness is consistently supported by experiments across both small and large-scale datasets.

Although, as suggested by Proposition 1, any set of random convolutional filters can be used to initialize attention maps, it is difficult to obtain both a clear attention map structure and sufficient

training flexibility from random non-impulse filters, especially when analyzed through low-rank approximation via singular value decomposition (SVD) and the Softmax operation. Thus, when aiming to initialize attention maps with a convolutional prior, impulse filters remain the most straightforward and robust choice.

## B  Proof for Proposition 1

Let $\mathbf{w} = [w_1, w_2, \ldots, w_D]^T \in \mathbb{R}^{D \times 1}$ be the channel mixing weights for one output channel and $\mathbf{H}_1, \mathbf{H}_2, \ldots, \mathbf{H}_D$ are the corresponding spatial convolution filters for each channel. Therefore, the result $\mathbf{y} \in \mathbb{R}^N$ after spatial and channel mixing can be represented as,

$$\mathbf{y} = \sum_{i=1}^{D} w_i \mathbf{H}_i \mathbf{x}_i, \tag{15}$$

With Remark 1, we can suppose the rank of $\mathbf{X} \approx \mathbf{ZA}$ is $k$, where $\mathbf{Z} = [\mathbf{z}_1, \ldots, z_k]$ and $k \ll D$, as illustrated in Fig. 2. We then obtain

$$\mathbf{y} \approx \sum_{i=1}^{D} \sum_{j=1}^{k} w_i a_{ji} \mathbf{H}_i \mathbf{z}_j = \sum_{j=1}^{k} \tilde{\mathbf{H}}_j \mathbf{z}_j, \tag{16}$$

where $a_{ji}$ refers to the row $j$, column $i$ element of $\mathbf{A}$, and $\tilde{\mathbf{H}}_j = \sum_{i=1}^{D} w_i a_{ji} \mathbf{H}_i$.

Remember that a linear combination of $f^2$ linearly independent filters $\mathbf{h}$ can express any arbitrary filter in filter space $\mathbb{R}^{f \times f}$, where $\mathbf{h}$ serves as the basis. Consequently, any desired $\tilde{\mathbf{H}}_1, \tilde{\mathbf{H}}_2, \ldots, \tilde{\mathbf{H}}_D$ can be achieved by only learning the channel mixing weights $\mathbf{w}$. Therefore, we obtain the following proposition.

## C  Convolutional Represetation Matrix

In Sec. 3, we interchangeably use the terms convolution filter $\mathbf{h}$ and convolution matrix $\mathbf{H}$. Additionally, we represent the impulse filter as a convolutional matrix. Here, we offer a detailed explanation of the relationship between the convolutional filters and the convolutional matrices.

Let us define a 2D convolution filter as $\mathbf{h} \in \mathbb{R}^{f \times f}$ with elements

$$\mathbf{h} = \begin{pmatrix} h_{11} & \cdots & h_{1f} \\ \vdots & \ddots & \vdots \\ h_{f1} & \cdots & h_{ff} \end{pmatrix}. \tag{17}$$

When $\mathbf{h}$ is convolved with an image $\mathbf{x} \in \mathbb{R}^{H \times W}$, this convolution operation is equivalent to a matrix multiplication

$$\mathrm{vec}(\mathbf{h} * \mathbf{x}) = \mathbf{H}\,\mathrm{vec}(\mathbf{x}), \tag{18}$$

where $\mathbf{H}$ is composed from the elements in $\mathbf{h}$ and zeros in the following format:

$$\mathbf{H} = \begin{pmatrix} \mathbf{F_1} & \mathbf{F_2} & \cdots & \mathbf{F_f} & \mathbf{0} & \mathbf{0} & \cdots & \mathbf{0} \\ \mathbf{0} & \mathbf{F_1} & \mathbf{F_2} & \cdots & \mathbf{F_f} & \mathbf{0} & \cdots & \mathbf{0} \\ \vdots & \ddots & \ddots & \ddots & \ddots & \ddots & \ddots & \vdots \\ \mathbf{0} & \cdots & \mathbf{0} & \mathbf{F_1} & \mathbf{F_2} & \cdots & \mathbf{F_f} & \mathbf{0} \\ \mathbf{0} & \cdots & \mathbf{0} & \mathbf{0} & \mathbf{F_1} & \mathbf{F_2} & \cdots & \mathbf{F_f} \end{pmatrix}, \tag{19}$$

where

$$\mathbf{F_i} = \begin{pmatrix} h_{i1} & h_{i2} & \cdots & h_{if} & 0 & 0 & \cdots & 0 \\ 0 & h_{i1} & h_{i2} & \cdots & h_{if} & 0 & \cdots & 0 \\ \vdots & \ddots & \ddots & \ddots & \ddots & \ddots & \ddots & \vdots \\ 0 & \cdots & 0 & h_{i1} & h_{i2} & \cdots & h_{if} & 0 \\ 0 & \cdots & 0 & 0 & h_{i1} & h_{i2} & \cdots & h_{if} \end{pmatrix}, \tag{20}$$

for $i = 1, 2, \ldots, f$. $\mathbf{F_i}$s are circulant matrices and $\mathbf{H}$ is a block circulant matrix with circulant block (BCCB). Note that convolutions may employ various padding strategies, but the circulant structure remains consistent. Here, we show the convolution matrix without any padding as an example.

Table 5: Ablation study on training settings of ViT-Base on ImageNet-1K dataset. Note that we compare the performance to the default initialization method (shaded in gray).

| Scaled LR | Repeated Augment | Default | Mimetic | | Ours (impulse) | |
|---|---|---|---|---|---|---|
| 7e-4 | 1.0 | 76.25 | 79.12 | 2.87↑ | **80.48** | **4.23**↑ |
| 1e-3 | 1.0 | 80.09 | 80.17 | 0.08↑ | **81.55** | **1.07**↑ |
| 1e-3 | 3.0 | 81.24 | 80.56 | 0.68↓ | **81.83** | **0.59**↑ |

# D  Pseudo Code for Solving $\mathbf{Q}_{\text{init}}$ and $\mathbf{K}_{\text{init}}$

---

**Algorithm 1** Convolutional Structured Impulse Initialization for ViT

---

**Input: P** $\qquad\qquad\qquad\qquad\qquad\qquad\qquad\qquad\qquad$ ▷ Input positional encoding
**Input:** $d, f, \alpha, \beta, \gamma$ $\qquad\qquad\qquad\qquad\qquad\qquad\qquad\qquad$ ▷ Hyperparameters
**Output: $\mathbf{Q}_{\text{init}}, \mathbf{K}_{\text{init}}$** $\qquad\qquad\qquad\qquad\qquad$ ▷ Initialized attention parameters
1: $N, D \leftarrow \text{shape}(\mathbf{P})$
2: $\mathbf{H}_{\text{impulse}} \leftarrow \text{ImpulseConvMatrix}(N, f)$ $\qquad$ ▷ Build 2D impulse convolution matrix
3: $\widetilde{\mathbf{M}} \leftarrow \alpha \mathbf{H}_{\text{impulse}} + \beta \mathbf{Z}$ $\qquad\qquad\qquad\qquad\qquad$ ▷ Get ideal map
4: $\widetilde{\mathbf{X}} \leftarrow \text{LayerNorm}(\mathbf{P})$ $\qquad\qquad\qquad\qquad\qquad$ ▷ Get pseudo input
5: $\mathbf{P}_{\text{inv}} \leftarrow (\widetilde{\mathbf{X}}^\top \widetilde{\mathbf{X}})^{-1} \widetilde{\mathbf{X}}$ $\qquad\qquad\qquad\qquad$ ▷ Get pseudo inverse of $\mathbf{P}$
6: $\hat{\mathbf{M}} \leftarrow \mathbf{P}_{\text{inv}} \widetilde{\mathbf{M}} \mathbf{P}_{\text{inv}}^\top$ $\qquad\qquad\qquad\qquad\qquad$ ▷ Get patterns $\mathbf{Q}\mathbf{K}^T$
7: $\mathbf{U}, \mathbf{s}, \mathbf{V}^\top \leftarrow \text{SVD}(\hat{\mathbf{M}})$ $\qquad\qquad\qquad\qquad\qquad$ ▷ SVD of $\hat{\mathbf{M}}$
8: $\widetilde{\mathbf{Q}}_{\text{init}} \leftarrow \mathbf{U}\sqrt{\mathbf{s}}, \quad \widetilde{\mathbf{K}}_{\text{init}} \leftarrow \mathbf{V}\sqrt{\mathbf{s}}$
9: $\widetilde{\mathbf{Q}}_{\text{init}} \leftarrow \widetilde{\mathbf{Q}}_{\text{init}}[:, :d], \quad \widetilde{\mathbf{K}}_{\text{init}} \leftarrow \widetilde{\mathbf{K}}_{\text{init}}[:, :d]$
10: $\widetilde{\mathbf{Q}}_{\text{init}} \leftarrow \widetilde{\mathbf{Q}}_{\text{init}}/\text{norm}(\mathbf{Q}), \quad \widetilde{\mathbf{K}}_{\text{init}} \leftarrow \widetilde{\mathbf{K}}_{\text{init}}/\text{norm}(\mathbf{K})$
11: $\mathbf{Q}_{\text{init}} \leftarrow \gamma \widetilde{\mathbf{Q}}_{\text{init}}, \quad \mathbf{K}_{\text{init}} \leftarrow \gamma \widetilde{\mathbf{K}}_{\text{init}}$
12: **return** $\mathbf{Q}_{\text{init}}, \mathbf{K}_{\text{init}}$

---

# E  Training Details and Training Curves for ViT-Base

At first, we found it challenging to reproduce the ViT-Base performance reported in DeiT [26], even when strictly following their specified training settings. Upon a careful examination of code differences across various versions of the timm [31] library, we identified two critical discrepancies that likely contributed to the performance gap: **(a) Learning rate scaling strategy:** In DeiT, the learning rate is linearly scaled with respect to the batch size: $\text{lr}_{\text{scaled}} = \text{lr}_{\text{base}} \times \frac{\text{batch size}}{512}$. In contrast, the current version of timm uses a square root scaling rule as the default for the AdamW optimizer: $\text{lr}_{\text{scaled}} = \text{lr}_{\text{base}} \times \sqrt{\frac{\text{batch size}}{512}}$. This discrepancy in the default setting leads to different effective learning rates, even when all the other hyperparameters are identical, and can substantially affect performance. **(b) Repeated data augmentation setting:** DeiT emphasizes the importance of the repeated data augmentation strategy, stating a drop in top-1 accuracy from $81.8\%$ to $76.5\%$ when it is disabled. However, they did not specify the exact augmentation weighting value in their original paper. After inspecting their specific version of code, we discovered that the default value for the repeated data augmentation was set to 3, whereas we used 1 in our main experiments, which may partially explain the performance discrepancy. We also aligned several other minor settings, such as the minimum learning rate during warm-up and at the end of training, as well as an additional 10 epochs for cool-down. However, we believe that these factors have only a marginal effect on the final performance, while the two aforementioned reasons remain the primary contributors to the observed discrepancy.

We present an ablation study on these two main settings difference in Tab. 5. In Fig. 5, we also present the training accuracy curves across epochs for different initialization methods with three different training configurations. Our impulse initialization consistently outperforms the default initialization throughout the entire training process, with a particularly significant advantage in the final training stage. While the mimetic initialization shows relatively faster initial convergence, it

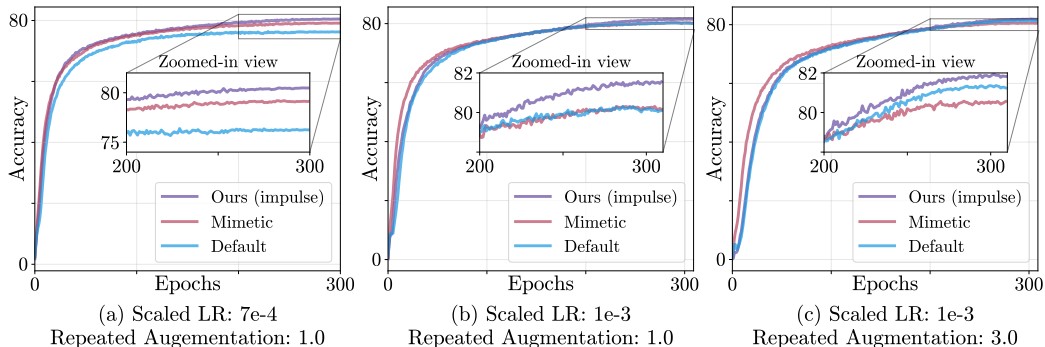

(a) Scaled LR: 7e-4
Repeated Augmentation: 1.0

(b) Scaled LR: 1e-3
Repeated Augmentation: 1.0

(c) Scaled LR: 1e-3
Repeated Augmentation: 3.0

Figure 5: Training curves of ViT-Base using default, mimetic, and impulse initialization under three different training configurations. The zoomed-in box shows the training curve in the final training stage from epoch 200 to epoch 300.

Table 6: Detailed scale of all the datasets used in the paper. ■ represents large-scale datasets, ▲ represents medium-scale datasets, and ● represents small-scale datasets. The datasets are ranked based on their scales following [33]. A '+' means we use both train and validation dataset in training. Note for Flowers and Pets datasets, we use both training and validation data for training.

| Dataset ↓ | #Classes | #Train images in each class | #Train images in total | #Images in total |
|---|---|---|---|---|
| ■ ImageNet-1K | 1K | $\sim 1$K | 1.3M | 1.4M |
| ▲ Food-101 | 101 | 750 | 75K | 101K |
| ▲ CIFAR-10 | 10 | 5K | 50K | 60K |
| ▲ CIFAR-100 | 100 | 500 | 50K | 60K |
| ● STL-10 | 10 | 500 | 5K | 13K |
| ● Flowers | 102 | 10+10 | 2K | 8K |
| ● Pets | 37 | 50+50 | 3.7K | 7.3K |

ultimately degrades performance in the final training stage. Our structured initialization method demonstrates robustness across different training configurations, consistently yielding over $80\%$ accuracy in all cases.

# F   Additional Results for Small and Medium-Scale Datasets

## F.1   Dataset Scale

For completeness, we provide the detailed dataset scales used in the main paper in Tab. 6. The ordering of the datasets follows that in [33].

## F.2   Additional Statistical Results

Here we provided the mean and standard deviation of 5 runs for ViT-Tiny with different initialization methods on small and medium-scale datasets in Fig. 6, which serves as additional statistical results of Tab. 1 in the main paper. Notably, our initialization method still outperforms other methods across small and medium-scale datasets. Especially for the smaller-scale datasets, our methods shows larger performance improvements, aligning with our findings in the main paper.

## F.3   Model Convergence Rate

We would like to clarify that while impulse filters mimic convolutional locality, they do not reuse weights like CNNs do—which may explain the fast convergence of CNNs. Interestingly, we did observe a faster convergence when using the mimetic or impulse initialization to replace the vanilla ViT model. We give an example of the training accuracy of ViT-Tiny with different initialization methods on CIFAR-10 during the optimization stage, as shown in Tab. 7. We can clearly see a faster

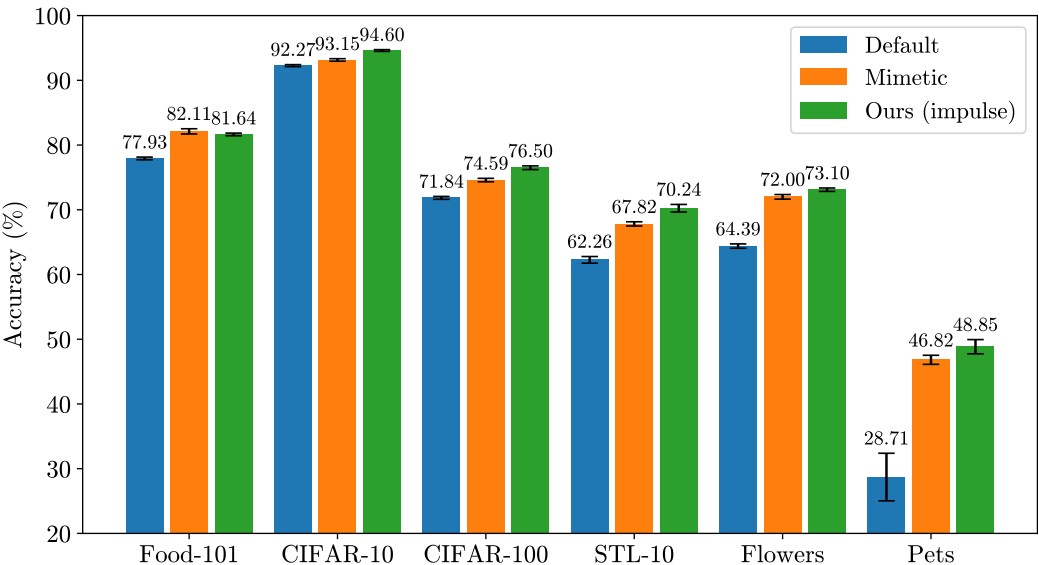

Figure 6: Classification accuracy of ViT-Tiny with different initialization methods on different datasets (mean of 5 runs). Our method consistently outperforms the default and the mimetic initializations. Except for the smallest-scale dataset Pets, all the methods show robust performance across different datasets with small error bars. Note that from left to right, the dataset scale decreases.

Table 7: Comparison of performance when training ViT-Tiny on CIFAR-10 at different epochs. Our method consistently achieves higher accuracy than both default and mimetic initialization baselines.

| Epoch | Default | Mimetic | Ours (Impulse) |
|-------|---------|---------|----------------|
| 10    | 47.95   | 49.05   | **54.62**      |
| 50    | 61.51   | 63.62   | **69.51**      |
| 100   | 76.87   | 77.64   | **81.87**      |

convergence of mimetic and our impulse initialization methods compared to the default initialization used in vanilla ViTs. One intuitive explanation is that the mimetic initialization yields $\sim 4\times$ the norm values of the default initialization, while our initialization method has $\sim 2\times$ the norm values of the default initialization. Different norm values will affect the gradient magnitudes and early training dynamics.

### F.4 Performance Improvement Increases as the Model Size Increases.

In the main paper Tab. 2, we also observe that as the model scale increase (*i.e.*, the number of heads increases), our method becomes relatively more expressive. Theoretically, as explained in Sec. 3, to replace the spatial kernels with fixed filters, the initialized attention heads must span the spatial filter basis. Therefore, for a $3\times3$ filter used in the experiments, this requires at least 9 independent heads to span the kernel space. Please note that in ConvMixer, each channel uses an independent filter, which satisfies this criterion. However, ViT models share the same filters within each head, which need a separate discussion: 1) ViT-Tiny has 3 heads, ViT-Small has 6 heads—insufficient to span to the kernel space; 2) ViT-Base has 12 heads—sufficient for replacing the spatial kernels. This analysis fully supports this performance improvement increase observation. However, as the motivation of the paper also suggested, training even larger ViT models on small datasets is notoriously challenging due to optimization instability and is out of the scope of this paper.

## G   Experimental Validation of the Propositions and Corollaries in Sec. 3

In this section, we conduct a series of preliminary experiments on CIFAR-10 to validate the propositions and corollaries in Sec. 3 of the main paper, with particular focus on demonstrating that in

Table 8: Classification accuracy(%) of ConvMixer (depth 8) with different filter sizes, embedding dimensions on CIFAR-10.

| Kernel Size | Embedding Dimension = 256 | | | | Embedding Dimension = 512 | | | |
|---|---|---|---|---|---|---|---|---|
| | Trained | Random | Impulse | Box | Trained | Random | Impulse | Box |
| 3 | 91.76 | 90.72 | 90.68 | 81.70 | 92.82 | 92.15 | 92.20 | 81.90 |
| 5 | 92.69 | 90.87 | 90.41 | 80.57 | 93.90 | 92.72 | 91.91 | 81.19 |
| 8 | 92.34 | 88.12 | 87.82 | 78.95 | 92.96 | 90.09 | 89.61 | 80.10 |

ConvMixer, fixed random impulse spatial filters can achieve comparable performance to learned filters. All experiments adhere to the training protocol proposed in [34], which is specifically tailored for evaluating diverse architectures on small-scale datasets such as CIFAR-10.

To support our theoretical findings in Sec. 3 concerning the effectiveness of random filters, we train ConvMixer [27] models with an embedding dimension of 256, a depth of 8, and a patch size of 2 on the CIFAR-10 dataset, using spatial filter sizes of 3, 5, and 8. We also include a variant with an embedding dimension of 512 to examine the impact of feature width. We evaluate the end-to-end trained ConvMixer alongside three initialization strategies: random (Corollary 1), impulse (Corollary 2), and box (Corollary 3). Note that in all three cases, only the spatial convolution filters are initialized—the models are evaluated without any training. The box filters use all-one values, effectively performing average pooling. The results are summarized in Tab. 8.

In conclusion, these experimental results reveal several key insights. First, comparing across columns (*i.e.*, initialization methods), both random and impulse initializations achieve performance comparable ($\geq 90\%$) to that of fully trained models, whereas the box initialization leads to significantly worse performance ($\sim 80\%$). This discrepancy can be attributed to the deficient rank of the box filters, which fail to span the full $f^2$-dimensional filter space, unlike random and impulse filters that are capable of forming a complete basis.

Second, when comparing across rows (*i.e.*, kernel sizes) with an embedding dimension of 256, the performance gap between trained and untrained (random or impulse) filters grows with kernel size, from $1\%$ (size 3) to $2\%$ (size 5) and $5\%$ (size 8). This occurs because larger kernels require more distinct filters to effectively span the filter space. However, the fixed embedding dimension constrains the number of such filters, reducing their ability to match the input rank as shown in Proposition 1. Notably, when the embedding dimension is doubled to 512, this performance gap narrows. In particular, for a kernel size of 3, the random and impulse initializations nearly match the performance of the trained filters, suggesting that sufficient embedding width compensates for the limitations of fixed filters.

# H  Attention Maps

Here we provide additional visualization of the attention maps for all 12 layers in Fig. 7. In particular, our structured initialization method offers different attention peaks on various heads, showing alignment with the impulse structures, while the mimetic initialization only presents main-diagonal peaks, and the default initialization shows little to no patterns. As stated in the main paper, both mimetic and default initialization methods use identical initialization for all attention heads.

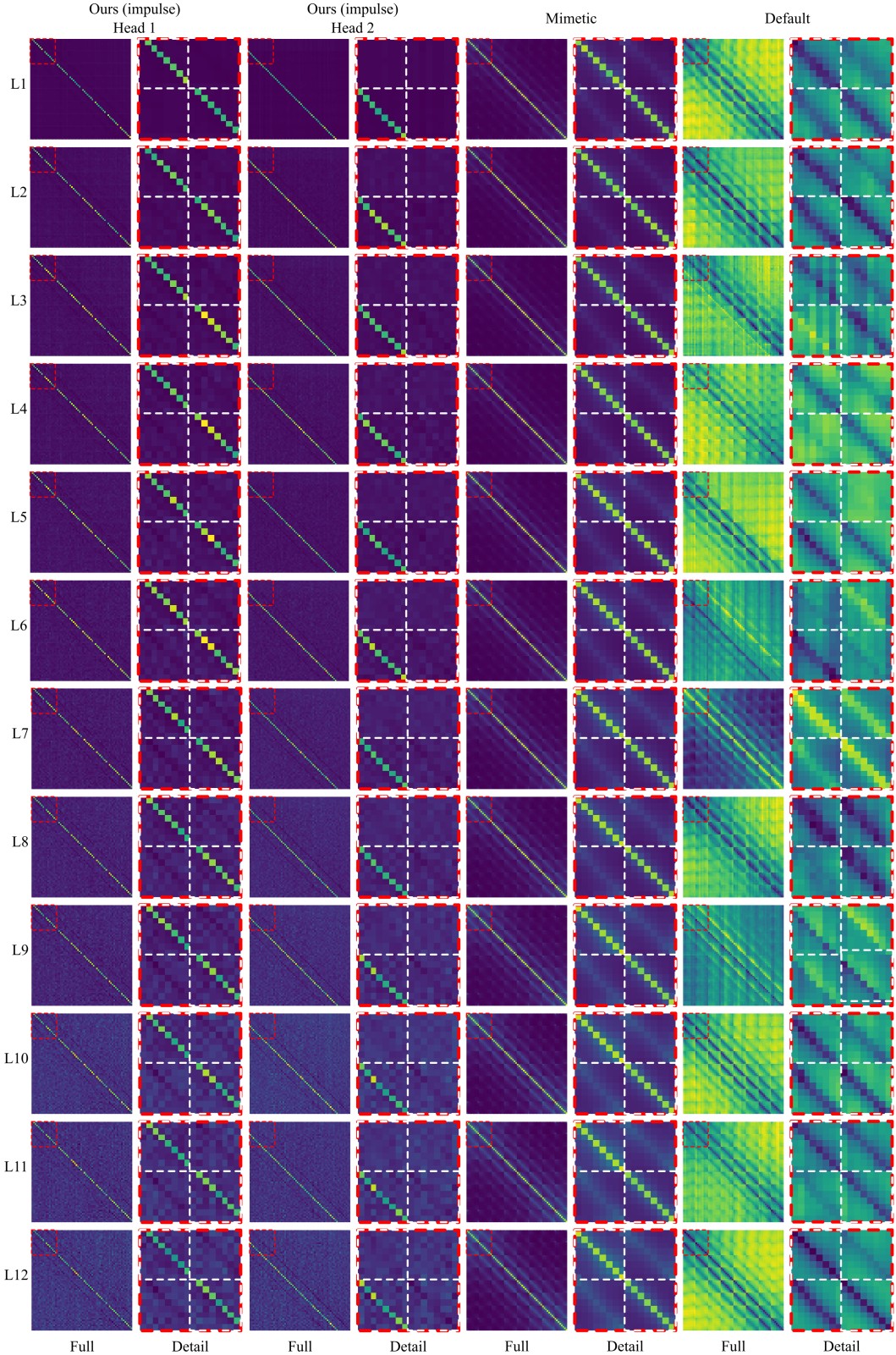

Figure 7: Visualization of attention maps in ViT-T using our impulse initialization method, mimetic [28], and default [31] initializations.

