# OpenReview forum: "Structured Initialization for Vision Transformers"
_NeurIPS.cc/2025/Conference — NeurIPS 2025 poster_

### Official Review · Reviewer_nTev · 2025-06-22

**Clarity:** 3
**Significance:** 3
**Originality:** 3
**Rating:** 5
**Confidence:** 4

**Summary:**

Vision Transformers enable global attention across image patches. However, in the vision domain, local patches tend to exhibit strong correlations, and ViTs struggle to capture such local dependencies effectively. As a result, ViTs often require more training time and larger training datasets, which poses challenges for training on small datasets. This paper introduces a structured initialization method that mimics the behavior of CNNs, helping ViTs better capture local correlations from the start. The proposed initialization leads to improved accuracy compared to standard random initialization and is particularly valuable when training ViTs on small datasets.

**Questions:**

- There is a lack of explanation regarding the visualization of the attention maps. The paper states that the proposed method is superior because it does not exhibit purely diagonal behavior, unlike the baselines. However, this point is not well elaborated. Could the authors clarify why non-diagonal attention patterns are beneficial?
- In Appendix B, the authors state that the proposed impulse initialization does not prevent the model from attending globally. However, the attention map visualizations suggest that the self-attention behavior in the proposed method is strongly influenced by the initialized impulse patterns. This appears to present a potential contradiction. Could the authors clarify how the model transitions from the localized, impulse-driven patterns at initialization to globally-aware attention during training?
- In the training pipeline, you mention that for the small datasets, “We follow the training recipe from [Initializing models with larger ones]”. However, that paper appears to rely on pretrained models for training. Could you clarify what specific aspects of their approach you adopted for the small datasets? Did you use pretrained models, or were there other elements of their training procedure that you followed?

**Ethical Concerns:**

["NO or VERY MINOR ethics concerns only"]

**Final Justification:**

While I remain somewhat unconvinced by the reasoning provided regarding the discrepancies in the DeiT-Base results, I feel that my main concerns have been addressed. The core idea of the paper is interesting, and the overall results are promising enough. I have increased my score accordingly.

**Limitations:**

yes

**Paper Formatting Concerns:**

--

**Quality:**

3

**Strengths And Weaknesses:**

Strengths:
- The paper is clearly written and well-organized.
- The proposed initialization is simple to implement and effective, without requiring architectural changes or pretrained weights.
- The method demonstrates consistent performance improvements across a wide range of datasets, particularly on small and medium-scale datasets.

Weaknesses:
- In Table 2, the reported accuracy for DeiT-Base is 76.25%, which is significantly lower than the 81.8% reported in the `timm` library. This 5–6% discrepancy suggests a potential issue with the experimental setup. The authors mention in the appendix that the gap is due to differences in learning rate scaling and data augmentation settings, and they report an improved accuracy of 81.24% for the default initialization under corrected settings. However, it is surprising that this corrected result is not included in Table 2 of the main paper. This omission is problematic, as the presented results show both the mimetic baseline and the proposed method underperform compared to random initialization from `timm`, which contradicts the paper’s stated goal of outperforming standard initialization. Moreover, the corrected accuracy (81.24%) still shows a ~0.5% gap from the 81.8% reference, indicating that the issue has not been fully resolved. I recommend the authors refer to the official DeiT repository for verified training hyperparameters: https://github.com/facebookresearch/deit/tree/main. Given that this paper targets a top-tier conference like NeurIPS, such discrepancie on such a well known baselines are not acceptable.
- While the paper claims broad applicability to Swin and MLP-Mixer models, the experimental evaluation for these architectures is very limited. For example, MLP-Mixer is evaluated only on CIFAR-10. If the authors wish to claim general applicability in the abstract and contributions, a more comprehensive evaluation is warranted.
- Since the use of random impulse filters may bias the Vision Transformer toward more local attention at the expense of global attention, the paper would benefit from including results on ImageNet variants such as ImageNet-V2, ImageNet-R, or ImageNet-Sketch to demonstrate robustness to distribution shifts.
- Since the proposed method mimics CNN-like locality, which typically leads to faster convergence, an ablation study on the number of training epochs would be insightful. It would be interesting to see whether the structured initialization leads to faster convergence compared to the baselines. For example, instead of training for 300 epochs, train for 100 epochs and compare the results... I would expect to get better results.
- The impulse filter generation is stochastic, so final performance may depend on the random seed. An evaluation of variance across different seeds would help assess the robustness of the method.
- The paper claims (line 36) that prior initialization methods are tied to specific architectures, but it does not specify which methods these are or explain how the limitation applies. This point needs further elaboration.
- The models are referred to as “ViT” even though they are trained using DeiT training recipes. To avoid confusion with other works (e.g., “How to Train Your ViT”), it would be more accurate to refer to these models as “DeiT.”

---

> ### Author Rebuttal · Authors · 2025-07-30
>
> We sincerely thank the reviewer for the constructive and thoughtful review. We're encouraged that the reviewer appreciates our method’s simplicity, clarity, and consistent improvements across small and medium-scale datasets. Below, we address each concern raised by the reviewer in detail.
>
> ## Weaknesses: ##
>
> ### **1. ImageNet-1K Performance discrepancy.**
>
> We kindly point the reviewer to **Appendix Section F** for a detailed explanation of this performance gap. In a word, different ```timm``` versions result in different training settings such as different augmentations, learning rate scalings, *etc* (summarized in **Table 5**). We will update the main paper with the detailed explanation and the updated results reported in the Appendix. In detail, we will update **Table 2** with the **Table 5** results. We will clearly report that under the training settings of a scaled learning rate of $1e{-}3$, and a repeated augmentation factor of $3.0$, with ***default initialization***, the vanilla ViT-Base achieves an accuracy of $81.24\%$ on ImageNet-1K dataset, while mimetic initialization achieves $80.56\%$, and our method achieves $81.83\%$.
>
> | Scaled LR | Repeated Augment | Default | Mimetic | Ours (impulse) |
> | :---: | :----: | :-----: | :----: | :----: |
> | 1e-3| 3.0 | 81.24 | 80.56, 0.68 $\downarrow$ | **81.83**, 0.59 $\uparrow$ |
>
> In addition, we would like to clarify that the remaining small gap (\~0.5\%) between our reported vanilla ViT result and the previous result is attributed to minor differences in hardware settings and implementation library versions.
> Nevertheless, both intuitive and quantitative experiments validate that no matter how the training configurations change, our method maintains the best performance with the biggest performance improvements over the vanilla ViT. These results indicate the effectiveness and the robustness of our method under different training configurations.
>
> ### **2. Evaluation on other ViT variants.**
>
> We would like to clarify that we include results on Swin and MLP-Mixer as a proof-of-concept application to demonstrate that our method does not require any structural modifications, and can be easily integrated into different architectures as a plug-in module. Further extensions to various non-ViT models could be an interesting future work.
>
> ### **3. More experiments on other larger-scale datasets, such as ImageNet variants.**
>
> We would like to emphasize that the focus of the paper is to introduce the architectural inductive bias from CNNs to initialize the attention map in Transformers to aid training on limited data settings. Although we have shown that our method maintains good performance on large-scale datasets such as ImageNet-1K, the focus is to validate the preservation of the transformer architectures and the flexibility of our method in learning dynamic global relationships when enough data is given—for example, training on different ImageNet variants—despite introducing the CNN inductive bias during initialization.
>
> ### **4. Model convergence rate.**
>
> We would like to clarify that while impulse filters mimic convolutional locality, they ***do not reuse weights*** like CNNs do—which may explain the fast convergence of CNNs. Interestingly, we did observe a faster convergence when using the mimetic or impulse initialization to replace the vanilla ViT model. We give an example of the training accuracy of ViT-Tiny with different initialization methods on CIFAR-10 during the optimization stage, as shown below:
>
> | | Default | Mimetic | Ours (impulse) |
> | :---: | :---: | :----: | :---: |
> | Epoch 10 | 47.95 | 49.05 | **54.62** |
> | Epoch 50 | 61.51 | 63.62 | **69.51** |
> | Epoch 100 | 76.87 | 77.64 | **81.87** |
>
> We can clearly see a faster convergence of mimetic and our impulse initialization methods compared to the default initialization used in vanilla ViTs.
> One intuitive explanation is that the mimetic initialization yields \~4x the norm values of the default initialization, while our initialization method has \~2x the norm values of the default initialization. Different norm values will affect the gradient magnitudes and early training dynamics. We will add such a discussion to the revised version.
>
> ### **5. Multiple runs with different random seeds and stochastic filters**
>
> We would like to refer the reviewer to **Appendix G.2** and **Figure 5** for detailed experiments on using different seeds with stochastic filter generations with **5** different runs. We will also update the results in the main paper accordingly.
>
> ### **6. Explain the limitation of previous methods: “their effectiveness is often tied to specific model architectures”**
>
> We would like to clarify that when we discuss the limitations of the previous methods, *“their effectiveness is often tied to specific model architectures”* refers to methods that are “architecture-dependent”—specialized to a certain model and cannot be treated as a “plug-in” module to generalize to other types of architectures. For example, methods like [14, 23, 24, 33] *“directly sampled weights from pretrained large-scale models as initialization for smaller models”* (**Line 75-76**), which means those weights are only effective for the specific models with specific parameter sizes. Unlike these previous methods, our approach is “architecture-agnostic”, does not require any pretrained weights, and can work as a “plug-in” module across different model architectures.
>
> ### **7. Naming of ViT and DeiT**
>
> We would like to clarify that we use conventional ViT architectures in our experiments, except for **Section 5.3**, where we adopt different ViT variants such as Swin Transformers and MLP-Mixer. Specifically, we use the default ViT implementation from the ```timm``` library for most of our experiments. We only follow the training recipe of DeiT for the ImageNet-1K experiments for its clarity and fair comparison to the previous methods.
>
> ## **Questions:** ##
>
> ### **1. Further explanations for the attention map visualization.**
>
> We would like to refer to **Section 5.4**, especially **Line 277-280**, for a detailed explanation of non-diagonal peaks in the attention patterns. Specifically, different attention peak patterns shown in **Figure 3** correspond to different impulse patterns that are applied to different attention heads. We would also point the reviewer to **Section 3**, where we theoretically show that as long as the initial filters span the spatial filter space, they are functionally equivalent to distinct spatial filters in CNNs in terms of their representation ability.
> From these perspectives, each attention head should ideally serve as an independent basis vector, and different heads should be orthogonal to one another to have sufficient representation ability to their counterparts in CNNs.
>
> From **Figure 3**, we clearly see that our initialization generates different attention heads initialized with different patterns (better representation ability) while default and mimetic methods only initialize attention heads in the same way (no sufficient representation ability compared to spatial filters in CNNs).
> Our structured initialization ensures that the model starts with diverse, localized spatial priors while preserving the flexibility to learn global dependencies through training.
>
> ### **2. Clarification of attention map visualization and how the proposed method still enables attending global features during training**
>
> We would like to clarify that the attention map shown in **Figure 3** and **Appendix Figure 6** are the attention maps during initialization—before training. Therefore, the attention maps show clear and distinct patterns for different initialization methods. Since our method is a “plug-in” module that only changes the initialization method, no architectural modifications to the transformer structure are made. Therefore, training such ViTs still allows the model to update the attention weights $Q$, $K$, $V$, and enables the model to learn global, dynamic dependencies through optimization.
>
> Nonetheless, we have observed that although the attention patterns differ at the initialization stage, the attention map of ViTs using our initialization method gradually yields similar patterns as the conventional ViTs with default initialization. The final heat map of different images also validates that our method does not prevent the model from attending to global features.
>
> Due to the rebuttal limit, we cannot show such figures here. We will update the revised version with attention maps before, during, and after training, as well as attention heat maps of different inference images to better support our argument.
>
> ### **3. Clarification on using pretrained models**
>
> We would like to clarify that we ***did not use any pretrained models*** for our methods except for **Section 5.5**, where the pretrained models are explicitly evaluated. For all other smaller-scale experiments, we only follow the training configurations from [33] for a fair comparison, including batch size, learning rate, number of epochs, *etc*. Although [33] initializes the model weights from large pretrained models, we ***did not adopt any pretraining step for our comparisons***. In contrast, we apply default, mimetic, and our impulse initialization methods to ViT-Tiny models from scratch. We will make this implementation detail clearer in the revised version.

---

> > ### Comment · Reviewer_nTev · 2025-08-01
> >
> > Thank you for the detailed explanation. While I remain somewhat unconvinced by the reasoning provided regarding the discrepancies in the DeiT-Base results, I feel that my main concerns have been addressed. The core idea of the paper is interesting, and the overall results are promising enough. I have increased my score accordingly.

---

> > > ### Author Response · Authors · 2025-08-02
> > >
> > > We thank the reviewer for acknowledging that our rebuttal has solved the main concerns. We would like to clarify the discrepancy in the DeiT-Base result further, especially the “remaining small gap (\~0.5\%) between our reported vanilla ViT result and the previous result is attributed to minor differences in hardware settings and implementation library versions”.
> > > Upon careful re-investigation, we would like to clarify that “minor differences in hardware settings” refers to the difference in the number of GPUs and per-GPU batch size used in our paper. Please note that our implementation **follows the exact setup described in the original DeiT paper**. However, the DeiT GitHub repo has a different setting compared to the original paper. Specifically, we have listed the difference in hardware settings in both the DeiT paper and the DeiT GitHub repo as follows:
> > >
> > > || Number of GPUs | Global batch size | per-GPU batch size |
> > > | :---: | :----: | :----: | :----: |
> > > | **DeiT paper (our implementation)** | 8 | 1024 | 128 |
> > > | **DeiT gitHub repo** | 16 | 1024 | 64 |
> > >
> > > This difference in GPU settings, in turn, affects the **mixup** and other data augmentation methods that are sensitive to batch size. In addition, there also exists a difference in the exact GPU type we used in our paper and reported in the DeiT paper. We believe this further explanation could provide more insights on the small accuracy gap (\~0.5\%).

---

### Official Review · Reviewer_Cjkb · 2025-06-30

**Clarity:** 2
**Significance:** 2
**Originality:** 2
**Rating:** 4
**Confidence:** 4

**Summary:**

This paper introduces a novel initialization strategy for ViTs that leverages the inductive biases of CNNs to improve the training of ViTs on small-scale datasets. The core idea is to initialize the attention maps based on random impulse convolution kernels, using random impulse convolution filters to initialize the query (Q) and key (K) matrices, thereby incorporating the architectural bias of CNNs into the initialization process without altering the Transformer architecture. Both the theoretical and empirical justification demonstrate the technical soundness of the method, and extensive results on various datasets reflect the effectiveness.

**Questions:**

Please refer to the weaknesses section.

Minor-suggestions:
Line 63: missing a blank after [21, 15]
Limitations: although the authors discuss the limitations, it is better to include potential plans to address them.
Implementation details: more details about the experimental settings and hyper-parameters should be provided.

**Ethical Concerns:**

["NO or VERY MINOR ethics concerns only"]

**Final Justification:**

Most of my concerns are addressed in the authors' rebuttal, i raised my score to weak accept.

**Limitations:**

Yes

**Paper Formatting Concerns:**

N/A, no formatting concerns found.

**Quality:**

2

**Strengths And Weaknesses:**

**Paper strength**:
1. Combine CNN and ViT Advantages: The proposed method combines the inductive biases of CNNs with the flexibility of ViTs, unlike methods that rely on pre-trained models, this structured initialization does not require large-scale pre-training, making it accessible for tasks with limited data.
2. Easy-to-implement and Maintains Transformer Flexibility: The proposed structured initialization dose not change the architecture, which is somewhat easy to implement and preserving ViTs’ ability to learn dynamic, long-range global features.
3. Theoretical analysis and extensive evaluation: the theoretical elaboration provide fundamental insight and extensive results reflect the effectiveness of the proposed method.


**Paper weakness**:

1. About the weights of value (V) and projection layers: The current method focuses on the initialization of the Q and K matrices, neglecting the initialization of value weights (V) and the projection matrix. A more comprehensive initialization strategy that includes more components could potentially enhance the overall quality of ViTs, as well as provides more in-depth analysis.
2. Why only focusing on limited-data scenarios? I wonder the practical meaning of this method to enable ViTs to perform comparable with CNNs under limited data. In other words, Transformers are better for scaling and CNNs are more suitable in relatively limited data applications, since CNNs perform well under such setting, why this paper focuses such setting instead of exploring the scaling capability of ViTs under large-scale settings?
3. Integration with architectural design method: Since the proposed method requires no architectural modification, it is encouraged to involve methods with dedicated model designs to further testify the flexibility and compatibility of the method.
4. Multi-scale structural initialization : considering the importance of multi-scale and hierarchical nature of images, it is suggested to explore multi-scale structural initialization and evaluate if there are consistent performance gains.

**Justification**:
1. In my opion, CNNs are relatively more suitable when limited data is available and Transformers have better scalibility, this paper's motivattion to enable Transformers to be as effective as CNNs under such scenarios is not suffecient. The essentiality and timeliness of this research is somewhat limited.
2. The empirical evaluation could not comprehensively reflect the generalization and efficacy of the proposed method.
3. Could the proposed method be scaled for larger model parameters with more training data, this is doubt to me.
Therefore, I tend to vote for rejection.

---

> ### Author Rebuttal · Authors · 2025-07-30
>
> We thank the reviewer for giving a detailed evaluation of our work. We are glad that the reviewer acknowledged that our method is easy to implement and effective. We provide the detailed response below.
>
> ## **Weaknesses:** ##
>
> ### **1. Initialization of $V$ and projection matrices.**
>
> We would like to remind the reviewer that our proposed initialization method theoretically originated from the architecture of CNNs, while in the CNN structure, there is no reference to guide the initialization of value $V$—only $Q$ and $K$ are responsible for ***spatial mixing***. In addition, unlike previous methods that initialize $Q$, $K$, and $V$ separately, initializing the attention map itself ensures the incorporation of architectural inductive bias of CNNs. In contrast, the $V$ and projection matrices mainly perform ***channel mixing***, which requires a different theoretical basis and motivations, and is out of the scope of this paper. Therefore, we use the default initialization adopted in the conventional Transformer to initialize other components such as FNN, normalization layers, and projection layers.
>
> ### **2. Why focus on limited-data regimes?**
>
> We would like to point out to the reviewer that the motivation of our proposed initialization method is based on the fact that in practical scenarios, it is highly desirable to have a model that 1) works well on smaller datasets without any pretraining, 2) scales seamlessly to larger datasets when needed, thus can be quickly and cost-effectively evaluate their suitability for larger learning tasks. In addition, such models are more accessible to researchers, educators, and students with limited computing resources to more readily investigate and explore the applications of ViTs.
>
> Therefore, we focus on showing that pure transformers (*e.g.,* ViTs) without architectural change can perform well in such limited data settings, while preserving the flexibility of learning global and dynamic features. As shown in the results, our initialization method significantly improves the performance of vanilla ViTs on smaller datasets while maintaining good performance on larger datasets without harming the scalability of the transformer architecture (ViTs).
>
> ### **3. Integration with other specialized architectures**
>
> We would like to emphasize that one of the advantages of our method is that no architectural modifications are needed, thus preserving the flexibility of the ViT structure, enabling good performance on large-scale datasets with larger models. Additionally, we acknowledge that our initialization method is a general and flexible “plug-in” module, and can be integrated with more specialized ViT designs. For example, we have already shown in **Section 5.3** and **Table 3** with the performance comparison on Swin-Transformer and MLP-Mixer—our proposed method achieves the best performance compared to default or mimetic initialization methods without changing the model architectures.
>
> ### **4. Multi-scale structured initialization.**
>
> We acknowledge that multi-scale or hierarchical structures are important in vision models, and exploring multi-scale initialization could be an interesting future work. However, we would like to emphasize that the focus of our paper is to demonstrate the effectiveness of using pure initialization to aid ViT training on small-scale datasets.
>
> ## **Justification:** ##
>
> ### **1 and 2. Motivation and impact of our research, generalizability, and effectiveness of our method.**
>
> We would like to clarify that the motivation of our research work is reasonable, practical, and addresses real-world settings where 1) pretraining of the model is infeasible; 2) (training) data is limited; 3) model reuse is required across various model scales. We kindly refer the reviewer to the existing line of work **[7,9,14,15,21,23,24,28,29,32,33,35,37]** in studying how to enable training transformer models, especially ViTs, on smaller-scale datasets, which we have discussed in the introduction and related work sections of the paper, to better understand the motivations of our research.
>
> In the paper, we have demonstrated the consistently good performance of our proposed method in 6 diverse smaller-scale datasets with significant improvements over the vanilla ViT models, as well as strong performance on the larger-scale dataset ImageNet-1K to confirm the scalability and the robustness of our method. We believe all these experiments have validated the generalizability and the effectiveness of our proposed method.
>
> ### **3. Performance on even larger-scale models and datasets.**
>
> We would like to emphasize that the focus of the paper is to introduce the architectural inductive bias from CNNs to initialize the attention map in Transformers to aid training on limited data settings. Although we have shown that our method maintains good performance on large-scale datasets such as ImageNet-1K, the focus is to validate the preservation of the transformer architectures and the flexibility of our method in learning dynamic global relationships despite introducing the CNN inductive bias during initialization. We would like to clarify that our method is a “plug-in” module that can be easily integrated into larger-scale models. However, discussions of such applications are beyond the scope of this paper.
>
> ## **Minor suggestions and implementation details.**
>
> We will correct the formatting issues. We would like to point the reviewer to **Section 5** and **Appendix Section F** for detailed training configurations. We will expand the appendix with more implementation and training details.

---

> > ### Comment · Reviewer_Cjkb · 2025-08-02
> > **Official Comment by Reviewer Cjkb**
> >
> > I thank all the authors for their work in addressing my concerns during rebuttal, yet my concerns remain as following:
> >
> > **1. About the initialization of V and projection matrices.**
> >
> > First, the current version of the proposed method focuses on initializing ***Q*** and ***K*** matrices with CNN-like filters, and the authors claim that ***Q*** and ***K*** are responsible for ***spatial mixing*** and ***V*** and projection matrices mainly perform ***channel mixing***. However, in the softmax attention caculation, ***Q*** and ***K*** calculate their dot product as attention scores, indicating the correlation between differen input visual patches. These scores are then multipilied with ***V*** to aggregate their context information.   This process is ***not specifically*** about spatial mixing or channel mixing, but rather about capturing the relationships between different parts of the input data. Moreover, the representation dimention of ***Q***, ***K***, and ***V*** are usually set to be equal in common practice, e.g., 768 in the original ViT. I respectively agree that their CNN-like initialization might provide strctural guidance to the attention map, but there's no such evidence or any reference to my knowledge that indicating ***Q*** and ***K*** focus on  spatial mixing while ***V*** focus on channel mixing.
> >
> > Second, my initial concern about the initilization of ***V*** and ***projection matrices*** means that I wonder whther the performance could be further improved when initializing more components with their proposed strategy. However, the authors seem think that such exploration is out of their scope. I personally disagree as ***V*** and ***projection matrices*** could also **benifit from structural guidances** to learn better feature representations, thus improving the model performance.  Additionally, I notice that in limitations from the appendix, the authors themselves present their neglection of initializing ***V*** and ***projection matrices*** as a limitation, does this means that the initilization of ***V*** and ***projection matrices*** are ***in the scope*** of this study?
> >
> > **2. About their focus on limited-date regimes**
> >
> > The authors provide an explain about  why they focus on improving ViTs' performance under limited data regimes, but since CNNs already show strong performance under such settings and ViTs are better in scaling with tremendous pre-trained ViT-based models available,  focusing on limite-data settings could not fully reflect the effectiveness of their proposed method, as also raised by Reviewer 7orw and Vn6j.
> >
> >
> > **3. About the motivation and impact**
> >
> > First, as I have pointed earlier, CNNs excel in deliverring good performance and there are massive pre-trained ViT models available in the community for limited-data settings, making the ***technical contribution of this paper limited for the community***.
> >
> > Second, their claim about their results on the so-called large-scale dataset ImageNet-1K is not sufficient to convey their contribution on model scalability. Moreover, their results in Tab.2 indicate that the  performance improvement increases as model size increases and the authors attribute this to the increase in the number of heads. Does this means that the performance gains could be consistently increased with larger model size and more heads? If yes, why the authors claim that they maintain  **comparative** performance on large-scale datasets instead of further improving their performance with the proposed strategy? If no, what's the rational behind the performance gains?
> >
> > Third, in the rebuttal of Reviewer 7orw, the authors explained ```training on larger-scale datasets makes the optimization less sensitive to the initialization```. From this perspective, it seems that the author agree that their initialization strategy might make no difference under large-scale training, then why their method delivers consistent and substantial performance gains under their so-called large-scale datasets? This is confusing.
> >
> > Lastly, for these related literatures cited in the main text and highlighted in the rebuttal, most of them were published before 2022, when large-scale model pre-training was very challenging and domain-specific small-scale applications were important at that time. However, such settings are relatively outdated and could be better improved with the advances of foundation models.
> >
> > **4. More implementation details about hypter-parameters, training, settings**
> >
> > As also raised by Reviewer 7orw and nTev, some results remain unconvinced without  detailed implementation details about hypter-parameters, training, and other settings. I strongly suggest the authors to include these detailes.

---

> > > ### Author Response · Authors · 2025-08-04
> > > **Response to reviewer Cjkb**
> > >
> > > We thank the reviewer for the detailed comments. We will address the remaining concerns below.
> > >
> > > ### **1. Initialization of $V$ and projection matrices.**
> > >
> > > We appreciate the reviewer's insight on the roles of $Q$, $K$, and $V$. Our terminology of **spatia mixing** and **channel mixing** is based on the operational structure, specifically:
> > >
> > > - Given input $X \in \mathbb{R}^{N {\times} M}$ (with $N$ patches and $M$ features), multiplication of weight matrix to the input **from the left**—*e.g.*, $W_s X$, where $W_s \in \mathbb{R}^{N {\times} N}$—mixes across the **spatial** dimension.
> > >
> > > - Multiplication of weight matrix to the input **from the right**—*e.g.*, $X W_c$, where $W_c \in \mathbb{R}^{M {\times} M}$—mixes across the **channel/feature** dimension.
> > >
> > > Accordingly, attention weights ($QK^\top$) form an $N {\times} N$ matrix, which governs ***how patches interact***—referred to as **spatial mixing**, while $V$ and projection layers operate in the feature space. Thus, it is reasonable to focus on introducing spatial priors through $Q$ and $K$. Please also refer to Section 3 of our paper and paper [3] for a detailed explanation of spatial mixing and channel mixing.
> > >
> > > We agree that initializing $V$ and projection layers with structural priors could be a promising future work direction. However, our paper specifically focuses on introducing CNN inductive bias into the attention map—*i.e.*, spatial structure.
> > >
> > > ### **2. Our paper focuses on limited-data regimes.**
> > >
> > > We would like to emphasize that the goal of our method is to improve the performance of ViTs under limited data settings **without architectural changes**—a key feature making our method different from other CNN-transformer-fused methods, maintaining the scalability and the flexibility of the ViT structure.
> > >
> > > In this paper, we have shown:
> > >
> > > - Strong performance gains on smaller-scale datasets, where inductive bias matters most.
> > >
> > > - Comparable if not better results on large-scale datasets, such as ImageNet-1K, showing no loss of scalability or flexibility of using our initialization method.
> > >
> > > In summary, we believe that our work, as the first method to introduce the initialization strategy that explicitly structures attention maps in ViTs without changing the ViT structures, opens up the research direction for domain-specific initialization strategies under limited resources.
> > >
> > > ### **3. Motivation and impact of our research.**
> > >
> > > The core motivation of our paper is to incorporate inductive bias through initialization, not through architectural constraints. This strategy enables ViTs to behave more like CNNs when data is limited, and still maintain the flexible transformer structure when data is adequate.
> > >
> > > Regarding the performance scaling in Table 2: 1) The performance improvement increases with the increase in model size (*i.e.*, more heads), as more heads help span a richer filter space. Please also refer to our response to reviewer Vn6j for a detailed explanation. 2) The main purpose of including results on ImageNet-1K is to validate that using our initialization method does not hurt the performance of ViTs on larger-scale datasets.
> > >
> > > Regarding reviewer 7orw's comment: The reviewer 7orw’s question concerned **even larger-scale datasets**, and we noted that benefits from initialization may become less significant as data size increases—a common assumption which is reasonable for general learning methods. However, this assumption does not contradict our results; instead, it reinforces our motivation.
> > >
> > > Lastly, please refer to our paper reference, where this line of work ranges from 2020 to 2025.
> > >
> > > ### **4. Implementation details.**
> > >
> > > Here we list the summarized training setups as follows:
> > >
> > > - For smaller-scale datasets, we used hyperparameters such as batch size, learning rate, epochs, *etc.* from the “weight selection” paper [33].
> > > - For the larger-scale dataset ImageNet-1K, we follow the official DeiT training recipe.
> > >
> > > Please also refer to the main paper, the appendix, and the response to reviewers for detailed implementation and training setups. We will update these details in the revised paper to ensure clarity and reproducibility.

---

> > > > ### Comment · Reviewer_Cjkb · 2025-08-04
> > > > **Official Comment by Reviewer Cjkb**
> > > >
> > > > My main concerns have been addressed.  I will raise my score accordingly, more implementation details, and if any, include more recent related works in the next version of your paper.

---

### Official Review · Reviewer_Vn6j · 2025-07-02

**Clarity:** 3
**Significance:** 2
**Originality:** 3
**Rating:** 5
**Confidence:** 4

**Summary:**

The paper proposes structured initialization for Vision Transformers, in which the authors replace the usual random-weight or mimetic initializations of Q and K with carefully designed random impulse convolution filters that embed the locality bias of CNNs directly into the initial attention maps. They justify this choice theoretically by showing that, owing to the rank-deficiency of patch embeddings, a set of impulse kernels spans the same functional space as trained spatial filters, so learning can be delegated to the channel-mixing layers while the spatial bias is “baked in.” Empirically, this single-line initialization change boosts ViT-Tiny accuracy by 2~8 points on six small/medium benchmarks (Food-101, CIFAR-10/100, STL-10, Flowers, Pets), matches or betters baseline and mimetic schemes on ImageNet-1K, and transfers cleanly to Swin Transformer and MLP-Mixer variants.

**Questions:**

None.

**Ethical Concerns:**

["NO or VERY MINOR ethics concerns only"]

**Final Justification:**

Thanks to the authors for the detailed response to my concerns. As my questions have been satisfactorily addressed, I will accordingly increase my score.

**Limitations:**

None.

**Paper Formatting Concerns:**

None.

**Quality:**

3

**Strengths And Weaknesses:**

Pros:
1. The idea of injecting inductive bias purely through initialization is highly innovative. It elegantly sidesteps the trade-offs of hybrid architectures, preserving the model's full expressivity while providing a much-needed structural prior for training.
2. The experiments are well-designed, covering a wide range of dataset sizes, multiple architectures. The results are consistently and significantly positive.

Cons:
1. The authors commendably acknowledge that their ImageNet baseline performance is slightly lower than in the original papers - DeiT. While their focus on relative improvement is valid, it slightly tempers the impact of the absolute numbers reported.
2. In the large-scale dataset experiments shown in Table 2, as the model size increases, the performance improvement with impulse becomes more significant. Is there any detailed analysis on this? If the gain is solid, it would be worth a deeper investigation. Based on this observation, it may be valuable to try running ViT-base on the small/medium-scale datasets in Table 1 to see if even greater improvements can be achieved.
3. The work focuses exclusively on impulse filters. While justified by Corollary 2 and the feature selection argument, it is unclear if other simple structures (e.g., Gaussian kernels, DoG filters) could offer different and potentially useful biases.

---

> ### Author Rebuttal · Authors · 2025-07-30
>
> We sincerely thank the reviewer for their positive and constructive feedback. We appreciate the recognition of the novelty and effectiveness of our approach. Below, we respond to each concern that was raised by the reviewer.
>
> ## **Weaknesses:** ##
>
> ### **1. ImageNet-1K performance discrepancy.**
>
> We kindly point the reviewer to **Appendix Section F** for a detailed explanation of this performance gap. To summarize, we have not only provided a detailed explanation, but also provided two different experiments in **Table 5** and **Figure 4** for comparisons of using different training configurations. Both intuitive and quantitative experiments validate that, regardless of the changes in training configurations, our method maintains the best performance with the largest performance improvements over the vanilla ViT. These results indicate the effectiveness and robustness of our method under different training configurations. In addition, the reviewer has also acknowledged that the reported “relative improvements are valid”.
>
> ### **2. Performance improvement increases as model size increases.**
>
> In short, we would like to point out that the performance improvement increase is due to the increase in the number of heads. Theoretically, as explained in **Section 3**, to replace the spatial kernels with fixed filters, the initialized attention heads must span the spatial filter basis. Therefore, for a **3x3** filter used in the experiments, this requires at least **9** independent heads to span the kernel space.
> Please note that in ConvMixer, each channel uses an independent filter, which satisfies this criterion. However, ViT models share the same filters within each head, which need a separate discussion:
>
> - ViT-Tiny has **3** heads — insufficient to span to the kernel space;
> - ViT-Small has **6** heads — insufficient to span to the kernel space;
> - ViT-Base has **12** heads — sufficient for replacing the spatial kernels.
>
> This analysis fully supports the observation that as the model scale increases (*i.e.*, the number of heads increases), our method becomes more expressive and the performance improvement of our method increases. We will make this analysis clearer in the revised version.
>
> However, as the motivation of the paper also suggested, training even larger ViT models on small datasets is notoriously challenging due to optimization instability and is out of the scope of this paper. We believe that the results in **Table 1** have sufficiently validated the arguments of the paper that our proposed initialization method is effective for training transformers on limited data settings.
>
> ### **3. Using filters beyond impulse filters.**
>
> We acknowledge that other filters, such as Gaussian kernels or DoG filters, encode useful locality information and could perform equally well, as shown in our theoretical analysis, especially in **Corollary 2**. However, we use impulse filters in our method based on both practical and theoretical reasons as stated below.
>
> 1) **Practicality**: As discussed in **Section 4** “why using impulse filters” (**Lines 163–173**), attention maps are generated by the softmax function, which can only be positive and naturally leads to peaked distributions—closely aligned with the structure of impulse filters. Designing initialization schemes that yield Gaussian-structured attention maps is significantly more difficult due to the non-linear softmax operation.
>
> 2) **Theoretical simplicity**: Impulse filters provide a minimal and orthogonal basis to span the spatial domain, making them a simple, clear, and controllable way to introduce spatial bias while preserving the architectural flexibility of the model.
>
> In a word, the most straightforward and suitable choice is random impulse convolution filters.
> In future work, we are interested in exploring structured initialization with alternative attention mechanisms, which may better support the use of Gaussian-like or more complex filters.

---

### Official Review · Reviewer_7orw · 2025-07-03

**Clarity:** 3
**Significance:** 3
**Originality:** 3
**Rating:** 5
**Confidence:** 3

**Summary:**

The paper introduces a new initialisation method for non-CNN architectures, based on CNN like filters. They use impulse filters for this and provide theoretical justification. The initialization strategy differs from prior work by initializing different attention heads with different filters. On small-scale datasets, this initialization improves performance significantly, similar to architectural inductive biases. However, unlike architectural inductive biases, they don't limit scalability on larger scale datasets (up to IN-1K).

**Questions:**

- Are there downsides to the structured initialisation?
- Would any benefit of initialisation for fine-tuning a pre-trained model remain when pre-training is scaled up?
- The paper claims better robustness to hyperparameter selection, but doesn't show it in the main paper. Maybe it's better to include it in the paper as it's part of the main message (including the conclusion)?
- Can we be sure that the initialization doesn't limit performance compared to random initialization when training on even larger scale datasets than IN1K? Or different types of training, such as self-supervised objectives. I think it's likely, but it's not sure.

**Ethical Concerns:**

["NO or VERY MINOR ethics concerns only"]

**Final Justification:**

I will maintain my score of accept because the paper is technically solid, introduces a novel initialization method with strong empirical results, and provides valuable insights into the role and impact of initialization. While the practical utility of the approach remains debatable, the contribution is likely relevant and informative for the community.

**Limitations:**

yes

**Paper Formatting Concerns:**

-

**Quality:**

3

**Strengths And Weaknesses:**

Strengths:
- The paper is clearly written.
- Inducing inductive bias through initialization rather than the architecture is an interesting and promising direction, as it benefits from the best of both worlds: good on small-scale without limiting large-scale.
- The paper finds theoretical justification for findings in recent work that show for ConvMixer like architecture, the spatial mixing doesn't have to be trained to achieve good performance.
- The paper shows their initialization even outperforms prior weight selection methods which use pre-trained models.
- The pretraining results are promising, highlighting that the benefit of the initialization remains even after pre-training.

Weaknesses:
- Though the paper shows they can improve ViT on small-scale datasets, they don't provide a comparison with ConvMixer (or similar). Thus it's unclear whether their method makes ViT actually useful on a small-scale, as one otherwise might still better use a CNN.
- DeiT-B 224 without distillation shows 81.4 accuracy on ImageNet-1K. In your reproduction it's 76.25. What explains this difference? It's a bit concerning to me, as there might be a problem in the training pipeline, which might change the conclusion, though unlikely.
- The paper argues this initialization is useful in domain-specific use-cases, as there limited data is available. However, this is never verified. Though experiments are done on small scale generic datasets, no pre-training is used there (except for one table with IN1K pre-training, which is arguable too limited). Therefore, it seems to me important, that it is verified that current off-the-shelf large-scale pre-trainings are indeed less effective on domain-specific data (which is dissimilar from generic pre-training datasets), and structured initialisation is more effective. I do understand this might be challenging and don't think this is crucial for this paper to be accepted.

---

> ### Author Rebuttal · Authors · 2025-07-30
>
> We sincerely thank the reviewer for the constructive and thoughtful comments. We are happy to know that our paper is well written, and the initialization idea is interesting, and has great theoretical and quantitative justifications. We will address each concern in detail.
>
> ## **Weaknesses:** ##
>
> ### **1. Comparison with CNNs (** ***e.g.*** **, ConvMixer) on small-scale datasets**
>
> We would like to clarify that it is difficult to give a definite and general comparison over whether CNNs or Transformers are better in performance because of different model architectures, training configurations, application scenarios, *etc*. However, to give an intuitive comparison and to validate the effectiveness of our method in aiding training ViTs on smaller-scale datasets, we did a simple experiment to compare ConvMixer and ViT. Specifically, we design 1) ConvMixer with depth=8, embedding dimension=256, and convolutional filter size=3; 2) a ViT with depth=8, embedding dimension=256, and number of heads=16. We show the accuracy on CIFAR-10 below:
>
> | Embedding Dimension | ConvMixer (trained) | ViT (default) | ViT (mimetic) | ViT (impulse) |
> | :---: | :----: | :----: | :----: | :---: |
> | $256$ | $\mathbf{91.76}$ | $\underline{84.62}$ | $88.97$ | $\mathbf{\underline{90.59}}$ |
>
> In this table, we show that the baseline ConvMixer with trained filters achieves the best result. When trained with vanilla ViT with default initialization, the performance drops **\~7\%**. Using mimetic initialization improves the accuracy by **\~4.4\%**, while using our impulse initialization method significantly improves the accuracy by **\~6\%**, making the performance of ViT (**90.59**) comparable with ConvMixer (**91.76**).
> Please note that for a fair comparison, we force the ViT to have the same depth, size, and embedding dimension. However, in practice, ViTs have different model configurations. For example, a ViT-Tiny model typically has 3 heads, 12 layers with an embedding dimension of 192.
>
> Nonetheless, this table demonstrates the effectiveness of our structured initialization in introducing CNN inductive bias into the ViTs and improving the performance under the limited data settings, making the accuracy comparable to ConvMixer. Meanwhile, having such a ViT structure unchanged enables the scalability and flexibility to learn more complex features. We will update these intuitive results to the revised version.
>
> ### **2. ImageNet-1K performance discrepancy explanation**
>
> We kindly point the reviewer to **Appendix Section F** for a detailed explanation of this performance gap. To summarize, we have not only provided a detailed explanation, but also provided two different experiments in **Table 5** and **Figure 4** for comparisons of using different training configurations. Both intuitive and quantitative experiments validate that no matter how the training configurations change, our method maintains the best performance with the biggest performance improvements over the vanilla ViT. These results indicate the effectiveness and the robustness of our method under different training configurations. We will update the main paper with the updated results reported in the Appendix.
>
> ### **3. Verifying the effectiveness of the proposed method in domain-specific use cases**
>
> We would like to clarify that we have already verified that using our initialization method is effective in domain-specific use cases where only limited data is present. Our paper focuses on showing that integrating the domain-specific prior knowledge, *i.e.*, CNN locality,—which is important in visual feature learning—into the large vision model, *i.e.*, ViTs as a structured attention map during initialization is effective through extensive experiments on different limited data settings. Similar to visual priors, we argue that other domain-specific knowledge can also be integrated into the model architectures with the same analogy to aid training such a model with limited data.
> In addition, we have already shown in **Section 5.5** and **Table 4** that applying our initialization method in the pretraining stage can generate better pretrained weights for models that rely on pretraining on large-scale datasets (weight selection model [33]), making the performance comparable to such model that pretrained on even larger-scale datasets such as ImageNet-21K.
>
> ## **Questions:** ##
>
> ### **1. Downsides of the proposed method**
>
> We point out to the reviewer that we have included a detailed limitation analysis in **Appendix Section A** due to the page limit. We will make this section to the main paper for the final version.
>
> ### **2. Would any benefit of initialization remain when the pretrained model is scaled up?**
>
> We kindly refer the reviewer to **Section 5.5** and **Table 4** of the main paper for a detailed analysis of applying different initialization strategies on pretrained models. Please note, the pretrained model we used (weight selection [33]) is a ViT-Small model pretrained on the ImageNet-21K dataset. Our method preserves on-par if not better performance than the vanilla pretrained model on datasets with various scales. This experiment validates the benefit of our proposed initialization when the pretrained mode is scaled up.
>
> ### **3. Robustness to hyperparameter selections**
>
> We kindly refer the reviewer to **Appendix Section F** (**Table 5** and **Figure 4**) for a detailed comparison of how robust our method is to different training configurations, such as hyperparameter selections. We would like to emphasize that under different training configurations, our method maintains the best performance while achieving the biggest improvements over the vanilla ViT model.
>
> ### **4. Will the performance on larger-scale datasets or using self-supervised learning continue to be good?**
>
> We would like to emphasize that our method is a “plug-in” module that only changes the initialization and preserves the ability of the large model to learn complex relationships with adequate data. In the paper, we have already shown in **Section 5.2** and **Table 2** that when different scaled ViTs trained with a large-scale dataset ImageNet1K, our method still achieves the best performance compared to the default or the mimetic initialization methods. In addition, training on larger-scale datasets makes the optimization less sensitive to the initialization. Therefore, when trained on even larger-scale datasets or using self-supervised learning, the ability of the model to learn more complex features will not be limited, and the performance of the model will not be downgraded.

---

> > ### Comment · Reviewer_7orw · 2025-08-04
> >
> > Thanks to the authors for the elaborate response.
> >
> > Weakness 1 has been partially addressed. Why would one choose a ViT if ConvMixer is still better? I understand that theoretically the ViT has the scalability and flexibility to learn more complex features, but the results do not show a benefit of using a ViT here.
> >
> > Weakness 2 has been fully resolved.
> >
> > Weakness 3 has been partially addressed. Specifically, it is unclear why one would use impulse initialisation in a domain specific use case, if one can also use pre-trained weights instead. What practical scenario would make impulse initialisation useful, and what empirical results confirm this? In other words, is there a domain-specific use case where any off-the-shelf pre-trained weights are worse than impulse initialisation from scratch?

---

> > > ### Author Response · Authors · 2025-08-05
> > >
> > > We thank the reviewer for acknowledging the clarification of our rebuttal. We will further address the confusion here.
> > >
> > > ### **Weakness 1: Choice between ViT and ConvMixer.**
> > >
> > > We would like to clarify that we acknowledge that ConvMixers often outperform ViTs on small datasets because of their strong inductive bias. However, we would like to emphasize that the goal of our paper ***is not to replace CNNs in the small-scale dataset settings***, but to ***narrow the performance gap for ViTs*** while preserving their scalability and architectural flexibility. We believe that our work, as the first method to introduce the initialization strategy that explicitly structures attention maps in ViTs without changing the ViT structures, opens up the research direction for general, unified initialization strategies under limited resources. As also mentioned in [28], our research may also “enhance the understanding of the inner-workings of deep models and lead to cheaper training and better optima”.
> > > In addition, we would like to clarify that our goal is to show the benefit of using our initialization method in ViTs compared to vanilla ViTs using default initialization under the small-scale dataset settings, instead of showing the benefit of using ViTs over CNNs.
> > >
> > > ### **Weakness 3: Choice of impulse initialization over pretrained weights in domain-specific cases.**
> > >
> > > We kindly refer the reviewer to Section 5.5 and Table 4, where we show that our initialization strategy remains effective in the pretraining scenarios, achieving comparable performance to the pretrained model (pretrained on ImageNet-21K) with reduced pretraining data (only using ImageNet-1K data).
> > >
> > > In addition to these empirical results, our approach introduces a novel perspective—using structured initialization to encode domain-specific inductive biases into attention maps. For instance, prior knowledge such as **local spatial structure** (as in CNNs) and **graph connectivity** (motivated by links between attention and GNNs) can be effectively embedded through attention map initialization without modifying the ViT architecture. Therefore, in data-limited domains where large-scale pretraining is infeasible, or the biases are explicitly known but difficult to learn during pretraining, this type of structured initialization embeds specific domain priors into the model, providing a lightweight and effective alternative to pretrained weights.

---

> > > > ### Comment · Reviewer_7orw · 2025-08-05
> > > >
> > > > Thanks to the authors for their response. I have no further question regarding weakness 1. Regarding weakness 3, I was wondering if the authors can give an example of a data-limited domain where large-scale pretraining is infeasible. It could be the case that in almost any domain existing pretrained weights still provide some benefit over random initialisation, as the learned low-level shape detectors for example are very generic, and might be just as effective if not more than impulse initialisation.

---

> > > > > ### Author Response · Authors · 2025-08-07
> > > > > **Response to Reviewer 7orw**
> > > > >
> > > > > Thanks for the further discussion. We would like to clarify that we acknowledge that pretraining is prevailing and effective nowadays, especially for large models like ViTs. Using pretrained weights rather than random weights to initialize the model will indeed incorporate prior knowledge and improve the performance. Please also refer to Section 5.5 and Table 4 for pretraining experiments.
> > > > > However, we would like to point out that ***1)*** under limited data settings, large-scale training (from scratch) is infeasible due to a lack of prior knowledge; ***2)*** delicate domain adaptation is still required for general pretrained models to work well on domain-specific use cases. A specific example would be in **medical imaging**, general pretrained models usually present suboptimal performance due to differences in data sizes, image features, and task specifications, *etc.*, as noted in [1, 2, 3].
> > > > >
> > > > > Nevertheless, we would like to restate that the goal of our paper **is not to replace CNNs or transfer learning**, but rather to **offer a new strategy to initialize ViTs**, especially under the limited data settings where training is difficult. In addition, we would like to refer to the previous research [28] to further restate the goal of this line of research.
> > > > >
> > > > > In mimetic initialization [28], the authors pointed out:
> > > > > > Fundamentally, we seek to investigate the question proposed by Zhang *et al.* (2022) [4]: might some of the benefits of pretraining actually just be a result of it serving as a good initialization? Our approach is to attempt to find good initializations that do not involve pretraining to begin to explore this question.
> > > > >
> > > > > We hope this will provide further insights into our discussion.
> > > > >
> > > > > #### References
> > > > >
> > > > > [1] Raghu, Maithra, Chiyuan Zhang, Jon Kleinberg, and Samy Bengio. "Transfusion: Understanding transfer learning for medical imaging." Advances in neural information processing systems 32 (2019).
> > > > >
> > > > > [2] Matsoukas, Christos, Johan Fredin Haslum, Moein Sorkhei, Magnus Söderberg, and Kevin Smith. "What makes transfer learning work for medical images: Feature reuse & other factors." In Proceedings of the IEEE/CVF Conference on Computer Vision and Pattern Recognition, pp. 9225-9234. 2022.
> > > > >
> > > > > [3] Davila, Ana, Jacinto Colan, and Yasuhisa Hasegawa. "Comparison of fine-tuning strategies for transfer learning in medical image classification." Image and Vision Computing 146 (2024): 105012.
> > > > >
> > > > > [4] Zhang, Yi, Arturs Backurs, Sébastien Bubeck, Ronen Eldan, Suriya Gunasekar, and Tal Wagner. "Unveiling transformers with lego: a synthetic reasoning task." arXiv preprint arXiv:2206.04301 (2022).

---

> > > > > > ### Comment · Reviewer_7orw · 2025-08-08
> > > > > >
> > > > > > Thanks to the authors for their further clarification. I have no further questions. I do would like to point out that the referenced papers do not seem to indicate situations where random initialisation is actually better than generic pre-trained weights for medical imaging. It would be interesting to see whether impulse initialisation can actually outperform generic pre-trained weights. If it can, this makes impulse initialisation genuinely useful, otherwise the only usefulness is to answer the question from [28]. If my understanding is wrong, feel free to correct me here.

---

> > > > > > > ### Author Response · Authors · 2025-08-09
> > > > > > > **Response to reviewer 7orw**
> > > > > > >
> > > > > > > We appreciate the reviewer’s interest in comparing impulse initialization with pre-trained weights. However, we would like to clarify that such a performance comparison is **not** the central goal of this work---despite that we have already shown results of applying our method on pretrained models. We would like to clarify that our main contribution is to improve the *understanding* of how initialization affects ViTs, rather than to propose a method that necessarily outperforms pretraining models.
> > > > > > >
> > > > > > > Here, we clarify the usefulness of our work in three aspects.
> > > > > > >
> > > > > > > 1. **Performance in the pretraining setting**---As shown in Section 5.5 and Table 4, we have conducted experiments under a pretraining setting, showing our method still outperforms default initialization while achieving comparable performance compared to even larger pretrained models. While it could be interesting to evaluate our method on medical images and directly compare against generic pre-trained weights, this is not our primary focus.
> > > > > > >
> > > > > > > 2. **Understanding the role of initialization**---By referring to the mimetic initialization paper [28] and the question raised in [4], we highlight that initialization is not only about competing with pretraining methods. It is about uncovering why initialization matters and how initialization helps with model training. For example, the weight selection paper [33] selects pretrained weights from a larger model to initialize a smaller ViT. While such methods are more closely related to transfer learning and pretraining, they still contribute to the broader goal of finding better initialization strategies for ViTs. In our paper, we have also provided theoretical and empirical analysis to understand the role of weight initialization in the model.
> > > > > > >
> > > > > > > 3. **Incorporating structure prior into attention maps** – In vanilla ViTs, attention maps at initialization are computed from randomly initialized $Q$ and $K$ weights, so dependencies are learned entirely from scratch. We show that it is possible to embed prior knowledge/known dependencies (*e.g.*, CNN inductive bias, GNN connectivity, *etc*) into the initialization structure of attention maps without limiting the model’s ability to learn new dependencies/information from data. We believe this is the most important contribution of our work.
> > > > > > >
> > > > > > > We hope this discussion will provide more insights.

---

### Decision · Program_Chairs · 2025-09-17

**Decision:**

Accept (poster)

**Comment:**

The paper proposes a structured initialization scheme for vision transformers, approximating impulse filters. This initialization substantially improves the performance in small-data regimes over standard initialization and also over some recent work.

The results are fairly solid, with substantial improvements on several datasets and architectures.

The main concern is about the DeiT baseline on Imagenet-1k performing much worse than the original implementation. This has been discussed and the authors provided updated numbers with updated data augmentation and learning rate schedule - the proposed initialization still helps, although much less. This does not really affect the main conclusions of the paper, but investigating the proposed method at larger scale would certainly be a valuable follow-up.

The reviewers are all positive about the paper, the results are solid, so I recommend acceptance.